# PROVABLE FILTER PRUNING FOR EFFICIENT NEURAL NETWORKS

**Lucas Liebenwein**[*]
CSAIL, MIT
lucasl@mit.edu

**Cenk Baykal**[*]
CSAIL, MIT
baykal@mit.edu

**Harry Lang**
CSAIL, MIT
harry1@mit.edu

**Dan Feldman**
University of Haifa
dannyf.post@gmail.com

**Daniela Rus**
CSAIL, MIT
rus@csail.mit.edu

## ABSTRACT

We present a provable, sampling-based approach for generating compact Convolutional Neural Networks (CNNs) by identifying and removing redundant filters from an over-parameterized network. Our algorithm uses a small batch of input data points to assign a saliency score to each filter and constructs an importance sampling distribution where filters that highly affect the output are sampled with correspondingly high probability. In contrast to existing filter pruning approaches, our method is simultaneously data-informed, exhibits provable guarantees on the size and performance of the pruned network, and is widely applicable to varying network architectures and data sets. Our analytical bounds bridge the notions of compressibility and importance of network structures, which gives rise to a fully-automated procedure for identifying and preserving filters in layers that are essential to the network's performance. Our experimental evaluations on popular architectures and data sets show that our algorithm consistently generates sparser and more efficient models than those constructed by existing filter pruning approaches.

## 1 INTRODUCTION

Despite widespread empirical success, modern networks with millions of parameters require excessive amounts of memory and computational resources to store and conduct inference. These stringent requirements make it challenging and prohibitive to deploy large neural networks on resource-limited platforms. A popular approach to alleviate these practical concerns is to utilize a pruning algorithm to remove redundant parameters from the original, over-parameterized network. The objective of network pruning is to generate a sparse, efficient model that achieves minimal loss in predictive power relative to that of the original network.

A common practice to obtain small, efficient network architectures is to train an over-parameterized network, prune it by removing the least significant weights, and re-train the pruned network (Gale et al., 2019; Frankle & Carbin, 2019; Han et al., 2015; Baykal et al., 2019b). This prune-retrain cycle is often repeated iteratively until the network cannot be pruned any further without incurring a significant loss in predictive accuracy relative to that of the original model. The computational complexity of this iterative procedure depends greatly on the effectiveness of the pruning algorithm used in identifying and preserving the essential structures of the original network. To this end, a diverse set of smart pruning strategies have been proposed in order to generate compact, accurate neural network models in a computationally efficient way.

However, modern pruning approaches[1] are generally based on heuristics (Han et al., 2015; Ullrich et al., 2017; He et al., 2018; Luo et al., 2017; Li et al., 2016; Lee et al., 2019; Yu et al., 2017a) that lack guarantees on the size and performance of the pruned network, require cumbersome ablation studies (Li et al., 2016; He et al., 2018) or manual hyper-parameter tuning (Luo et al., 2017), or

---

[*]These authors contributed equally to this work.

[1]We refer the reader to Sec. A of the appendix for additional details about the related work.

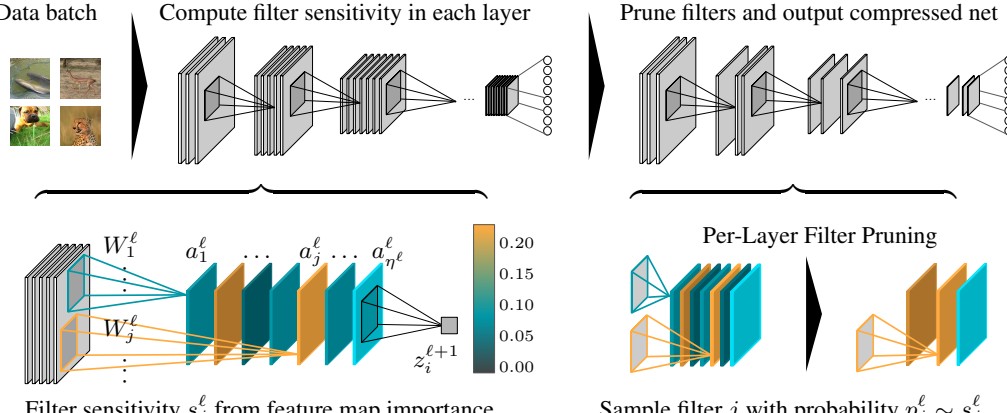

Figure 1: Overview of our pruning method. We use a small batch of data points to quantify the relative importance $s_j^\ell$ of each filter $W_j^\ell$ in layer $\ell$ by considering the importance of the corresponding feature map $a_j^\ell = \phi(z_j^\ell)$ in computing the output $z^{\ell+1}$ of layer $\ell + 1$, where $\phi(\cdot)$ is the non-linear activation function. We then prune filters by sampling each filter $j$ with probability proportional to $s_j^\ell$ and removing the filters that were not sampled. We invoke the filter pruning procedure each layer to obtain the pruned network (the *prune* step); we then retrain the pruned network (*retrain* step), and repeat the *prune-retrain* cycle iteratively.

heavily rely on assumptions such that parameters with large weight magnitudes are more important – which does not hold in general (Ye et al., 2018; Li et al., 2016; Yu et al., 2017a; Han et al., 2015).

In this paper, we introduce a data-informed algorithm for pruning redundant filters in Convolutional Neural Networks while incurring minimal loss in the network's accuracy (see Fig. 1 for an overview). At the heart of our method lies a novel definition of filter importance, i.e., filter *sensitivity*, that is computed by using a small batch of input points. We prove that by empirically evaluating the relative contribution of each filter to the output of the layer, we can accurately capture its importance with respect to the other filters in the network. We show that sampling filters with probabilities proportional to their sensitivities leads to an importance sampling scheme with low variance, which enables us to establish rigorous theoretical guarantees on the size and performance of the resulting pruned network. Our analysis helps bridge the notions of compressibility and importance of each network layer: layers that are more compressible are less important for preserving the output of the original network, and vice-versa. Hence, we obtain and introduce a fully-automated sample size allocation procedure for properly identifying and preserving critical network structures as a corollary.

Unlike weight pruning approaches that lead to irregular sparsity patterns – requiring specialized libraries or hardware to enable computational speedups – our approach compresses the original network to a slimmer subnetwork by pruning filters, which enables accelerated inference with any off-the-shelf deep learning library and hardware. We evaluate and compare the effectiveness of our approach in pruning a diverse set of network architectures trained on real-world data sets. Our empirical results show that our approach generates sparser and more efficient models with minimal loss in accuracy when compared to those generated by state-of-the-art filter pruning approaches.[2]

## 2 SAMPLING-BASED FILTER PRUNING

In this section, we introduce the network pruning problem and outline our sampling-based filter pruning procedure and its theoretical properties. We extend the notion of *empirical sensitivity* (Baykal et al., 2019a) to quantify the importance of each filter using a small set of input points. We show that our importance criterion enables us to construct a low-variance importance sampling distribution over the filters in each layer. We conclude by showing that our approach can eliminate a large fraction of filters while ensuring that the output of each layer is approximately preserved.

---

[2]Code available at `https://github.com/lucaslie/provable_pruning`

## 2.1 PRELIMINARIES

Consider a trained $L$ layer network with parameters $\theta = (W^1, \ldots, W^L)$, where $W^\ell$ denotes the 4-dimensional tensor in layer $\ell \in [L]$, $W_j^\ell$ filter $j \in [\eta^\ell]$, and $\eta^\ell$ the number of filters in layer $\ell$. Moreover, let $W_{:j}^{\ell+1}$ be channel $j$ of tensor $W^{\ell+1}$ that corresponds to filter $W_j^\ell$. We let $\mathcal{X} \subset \mathbb{R}^d$ and $\mathcal{Y} \subset \mathbb{R}^k$ denote the input and output space, respectively. The marginal distribution over the input space is given by $\mathcal{D}$. For an input $x \in \mathcal{X}$ to the network, we let $z^\ell(x)$ and $a^\ell(x) = \phi(z^\ell(x))$ denote the pre-activation and activation of layer $\ell$, where $\phi$ is the activation function (applied entry-wise). The $j^{\text{th}}$ feature map of layer $\ell$ is given by $a_j^\ell(x) = \phi(z_j^\ell(x))$ (see Fig. 1). For a given input $x \in \mathcal{X}$, the output of the neural network with parameters $\theta$ is given by $f_\theta(x)$.

Our overarching goal is to prune filters from each layer $\ell \in [L]$ by random sampling to generate a compact reparameterization of $\theta$, $\hat{\theta} = (\hat{W}^1, \ldots, \hat{W}^L)$, where the number of filters in the pruned weight tensor $\hat{W}^\ell$ is a small fraction of the number of filters in the original (uncompressed) tensor $W^\ell$. Let $\text{size}(\theta)$ denote the total number of parameters in the network, i.e., the sum of the number of weights over each $W^\ell \in (W^1, \ldots, W^L)$.

**Pruning Objective** For a given $\varepsilon, \delta \in (0, 1)$, our objective is to generate a compressed network with parameters $\hat{\theta}$ such that $\text{size}(\hat{\theta}) \ll \text{size}(\theta)$ and $\mathbb{P}_{x \sim \mathcal{D}, \hat{\theta}}(f_{\hat{\theta}}(x) \in (1 \pm \varepsilon) f_\theta(x)) \geq 1 - \delta$, where $f_{\hat{\theta}}(x) \in (1 \pm \varepsilon) f_\theta(x)$ denotes an entry-wise guarantee over the output neurons $f_{\hat{\theta}}(x), f_\theta(x) \in \mathcal{Y}$.

## 2.2 SAMPLING-BASED PRUNING

Our sampling-based filter pruning algorithm for an arbitrary layer $\ell \in [L]$ is depicted as Alg. 1. The sampling procedure takes as input the set of $\eta^\ell$ *channels* in layer $\ell + 1$ that constitute the weight tensor $W^{\ell+1}$, i.e., $W^{\ell+1} = [W_{:1}^{\ell+1}, \ldots, W_{:\eta^\ell}^{\ell+1}]$ as well as the desired relative error and failure probability, $\varepsilon, \delta \in (0, 1)$, respectively. In Line 2 we construct the importance sampling distribution over the feature maps corresponding to the channels by leveraging the *empirical sensitivity* of each feature map $j \in [\eta^\ell]$ as defined in (1) and explained in detail in the following subsections. Note that we initially prune *channels* from $W^{\ell+1}$, but as we prune channels from $W^{\ell+1}$ we can simultaneously prune the corresponding *filters* in $W^\ell$.

---

**Algorithm 1** PRUNECHANNELS$(W^{\ell+1}, \varepsilon, \delta, s^\ell)$

---

**Input:** $W^{\ell+1} = [W_{:1}^{\ell+1}, \ldots, W_{:\eta^\ell}^{\ell+1}]$: original channels; $\varepsilon$: relative error; $\delta$: failure probability; $s^\ell$: feature map sensitivities as in (1)

**Output:** $\hat{W}^{\ell+1}$: pruned channels

1: $S^\ell \leftarrow \sum_{j \in [\eta^\ell]} s_j^\ell$ {where $s_j^\ell$ is as in (1)}
2: $p_j^\ell \leftarrow s_j^\ell / S^\ell \quad \forall j \in [\eta^\ell]$
3: $m^\ell \leftarrow \lceil (6 + 2\varepsilon) S^\ell K \log(4\eta_*/\delta)\varepsilon^{-2} \rceil$
4: $\hat{W}^{\ell+1} \leftarrow [0, \ldots, 0]$ {same dimensions as $W^{\ell+1}$}
5: **for** $k \in [m^\ell]$ **do**
6: $\quad c(k) \leftarrow$ random draw from $p^\ell = (p_1^\ell, \ldots, p_{\eta^\ell}^\ell)$

7: $\quad \hat{W}_{:c(k)}^{\ell+1} \leftarrow \hat{W}_{:c(k)}^{\ell+1} + W_{:c(k)}^{\ell+1}/m^\ell p_{c(k)}^\ell$
8: **end for**
9: **return** $\hat{W}^{\ell+1} = [\hat{W}_{:1}^{\ell+1}, \ldots, \hat{W}_{:\eta^\ell}^{\ell+1}]$;

---

We subsequently set the sample complexity $m^\ell$ as a function of the given error ($\varepsilon$) and failure probability ($\delta$) parameters in order to ensure that, after the pruning (i.e., sampling) procedure, the *approximate* output – with respect to the sampled channels $\hat{W}^{\ell+1}$ – of the layer will approximate the *true* output of the layer – with respect to the original tensor – up to a multiplicative factor of $(1 \pm \varepsilon)$, with probability at least $1 - \delta$. Intuitively, more samples are required to achieve a low specified error $\varepsilon$ with low failure probability $\delta$, and vice-versa. We then proceed to sample $m^l$ times with replacement according to distribution $p^\ell$ ( Lines 5-8) and reweigh each sample by a factor that is inversely proportional to its sample probability to obtain an *unbiased* estimator for the layer's output (see below). The unsampled channels in $W^{\ell+1}$ – and the corresponding filters in $W^\ell$ – are subsequently discarded, leading to a reduction in the layer's size.

## 2.3 A TIGHTLY-CONCENTRATED ESTIMATOR

We now turn our attention to analyzing the influence of the sampled channels $\hat{W}^{\ell+1}$ (as in Alg. 1) on layer $\ell + 1$. For ease of presentation, we will henceforth assume that the layer is linear[3] and will omit

---

[3]The extension to CNNs follows directly as outlined Sec. B of the supplementary material.

explicit references to the input $x$ whenever appropriate. Note that the *true* pre-activation of layer $\ell+1$ is given by $z^{\ell+1} = W^{\ell+1}a^\ell$, and the *approximate* pre-activation with respect to $\hat{W}^{\ell+1}$ is given by $\hat{z}^{\ell+1} = \hat{W}^{\ell+1}a^\ell$. By construction of $\hat{W}^{\ell+1}$ in Alg. 1, we equivalently have for each entry $i \in [\eta^{\ell+1}]$

$$\hat{z}_i^{\ell+1} = \frac{1}{m}\sum_{k=1}^m Y_{ik}, \quad \text{where} \quad Y_{ik} = W_{ic(k)}^{\ell+1}\frac{a_{c(k)}^\ell}{p_{c(k)}^\ell}, c(k) \sim p \quad \forall k.$$

By reweighing our samples, we obtain an unbiased estimator for each entry $i$ of the true pre-activation output, i.e., $\mathbb{E}\left[\hat{z}_i^{\ell+1}\right] = z_i^{\ell+1}$ – which follows by the linearity of expectation and the fact that $\mathbb{E}\left[Y_{ik}\right] = z_i^{\ell+1}$ for each $k \in [m]$ –, and so we have for the entire vector $\mathbb{E}_{\hat{W}^{\ell+1}}[\hat{z}^{\ell+1}] = z^{\ell+1}$. So far, we have shown that in expectation, our channel sampling procedure incurs zero error owing to its unbiasedness. However, our objective is to obtain a high probability bound on the entry-wise deviation $\left|\hat{z}_i^{\ell+1} - z_i^{\ell+1}\right|$ for each entry $i$, which implies that we have to show that our estimator $\hat{z}_i^{\ell+1}$ is highly concentrated around its mean $z_i^{\ell+1}$. To do so, we leverage the following standard result.

**Theorem 1** (Bernstein's inequality (Vershynin, 2016)). *Let $Y_1, \ldots, Y_m$ be a sequence of $m$ i.i.d. random variables satisfying $\max_{k \in [m]} |Y_k - \mathbb{E}\left[Y_k\right]| \le R$, and let $Y = \sum_{k=1}^m Y_k$ denote their sum. Then, for every $\varepsilon \ge 0$, $\delta \in (0,1)$, we have that $\mathbb{P}\left(|Y/m - \mathbb{E}\left[Y_k\right]| \ge \varepsilon\,\mathbb{E}\left[Y_k\right]\right) \le \delta$ for*

$$m \ge \frac{\log(2/\delta)}{(\varepsilon E[Y_k])^2}\left(\mathrm{Var}(Y_k) + \frac{2}{3}\varepsilon\,\mathbb{E}\left[Y_k\right]R\right).$$

Letting $i \in [\eta^{\ell+1}]$ be arbitrary and applying Theorem 1 to the mean of the random variables $(Y_{ik})_{k \in [m]}$, i.e., to $\hat{z}_i^{\ell+1}$, we observe that the number of samples required for a sufficiently high concentration around the mean is highly dependent on the magnitude and variance of the random variables $(Y_{ik})_k$. By definition of $Y_{ik}$, observe that these expressions are explicit functions of the sampling distribution $p^\ell$. Thus, to minimize[4] the number of samples required to achieve high concentration we require a judiciously defined sampling distribution that simultaneously minimizes both $R_i$ and $\mathrm{Var}(Y_{ik})$. For example, the naive approach of uniform sampling, i.e., $p_j^\ell = 1/\eta^\ell$ for each $j \in [\eta^\ell]$ also leads to an unbiased estimator, however, for uniform sampling we have $\mathrm{Var}(Y_{ik}) \approx \eta^\ell\,\mathbb{E}\left[Y_{ik}\right]^2$ and $R_i \approx \eta^\ell \max_k(w_{ik}^{\ell+1}a_k^\ell)$ and so $\mathrm{Var}(Y_{ik}), R \in \Omega(\eta^\ell)$ in the general case, leading to a linear sampling complexity $m \in \Omega(\eta^\ell)$ by Theorem 1.

## 2.4 Empirical Sensitivity (ES)

To obtain a better sampling distribution, we extend the notion of Empirical Sensitivity (ES) introduced by Baykal et al. (2019a) to prune channels. Specifically, for $W^{\ell+1} \ge 0$ (the generalization can be found in Appendix B) we let the sensitivity $s_j^\ell$ of feature map $j$ in $\ell$ be defined as

$$s_j^\ell = \max_{x \in \mathcal{S}}\max_{i \in [\eta^{\ell+1}]}\frac{w_{ij}^{\ell+1}a_j^\ell(x)}{\sum_{k \in [\eta^\ell]}w_{ik}^{\ell+1}a_k^\ell(x)}, \tag{1}$$

where $\mathcal{S}$ is a set of $t$ independent and identically (i.i.d.) points drawn from $\mathcal{D}$. Intuitively, the sensitivity of feature map $j \in [\eta^\ell]$ is the maximum (over $i \in [\eta^{\ell+1}]$) relative impact that feature map $j$ had on any pre-activation in the next layer $z_i^{\ell+1}$. We then define the probability of sampling each channel as in Alg. 1: $j \in [\eta^\ell]$ as $p_j = s_j^\ell/S^\ell$, where $S^\ell = \sum_j s_j^\ell$ is the sum of sensitivities. Under a mild assumption on the distribution – that is satisfied by a wide class of distributions, such as the Uniform, Gaussian, Exponential, among others – of activations (Asm. 1 in Sec. B of the supplementary), ES enables us to leverage the inherent stochasticity in the draw $x \sim \mathcal{D}$ and establish (see Lemmas 5, 6, and 7 in Sec. B) that with high probability (over the randomness in $\mathcal{S}$ and $x$) that

$$\mathrm{Var}(Y_{ik}) \in \Theta(S\,\mathbb{E}\left[Y_{ik}\right]^2) \quad \text{and} \quad R \in \Theta(S\,\mathbb{E}\left[Y_{ik}\right]) \qquad \forall i \in [\eta^{\ell+1}]$$

and that the sampling complexity is given by $m \in \Theta(S\log(2/\delta)\,\varepsilon^{-2})$ by Theorem 1.

We note that ES does not require knowledge of the data distribution $\mathcal{D}$ and is easy to compute in practice by randomly drawing a small set of input points $\mathcal{S}$ from the validation set and passing

---

[4]We define the minimization with respect to sample complexity from Theorem 1, which serves as a sufficiently good proxy as Bernstein's inequality is tight up to logarithmic factors (Tropp et al., 2015).

the points in $\mathcal{S}$ through the network. This stands in contrast with the sensitivity framework used in state-of-the-art coresets constructions (Braverman et al., 2016; Bachem et al., 2017), where the sensitivity is defined to be with respect to the supremum over all $x \in \mathrm{supp}(\mathcal{D})$ in (1) instead of a maximum over $x \in \mathcal{S}$. As also noted by Baykal et al. (2019a), ES inherently considers data points that are likely to be drawn from the distribution $\mathcal{D}$ in practice, leading to a more practical and informed sampling distribution with lower sampling complexity.

Our insights from the discussion in this section culminate in the core theorem below (Thm. 2), which establishes that the pruned channels $\hat{W}^{\ell+1}$ (corresponding to pruned filters in $W^\ell$) generated by Alg. 1 is such that the output of layer $\ell + 1$ is well-approximated for each entry.

**Theorem 2.** *Let $\varepsilon, \delta \in (0, 1), \ell \in [L]$, and let $\mathcal{S}$ be a set of $\Theta(\log(\eta_*/\delta))$ i.i.d. samples drawn from $\mathcal{D}$. Then, $\hat{W}^{\ell+1}$ contains at most $\mathcal{O}(S^\ell \log(\eta_*/\delta)\varepsilon^{-2})$ channels and for $x \sim \mathcal{D}$, with probability at least $1 - \delta$, we have $\hat{z}^{\ell+1} \in (1 \pm \varepsilon)z^{\ell+1}$ (entry-wise), where $\eta_* = \max_{\ell \in [L]} \eta^\ell$.*

Theorem 2 can be generalized to hold for all weights and applied iteratively to obtain layer-wise approximation guarantees for the output of each layer. The resulting layer-wise error can then be propagated through the layers to obtain a guarantee on the final output of the compressed network. In particular, applying the error propagation bounds of Baykal et al. (2019a), we establish our main compression theorem below. The proofs and additional details can be found in the appendix (Sec. B).

**Theorem 3.** *Let $\varepsilon, \delta \in (0, 1)$ be arbitrary, let $\mathcal{S} \subset \mathcal{X}$ denote the set of $\lceil K' \log(4\eta/\delta) \rceil$ i.i.d. points drawn from $\mathcal{D}$, and suppose we are given a network with parameters $\theta = (W^1, \dots, W^L)$. Consider the set of parameters $\hat{\theta} = (\hat{W}^1, \dots, \hat{W}^L)$ generated by pruning channels of $\theta$ according to Alg. 2 for each $\ell \in [L]$. Then, $\hat{\theta}$ satisfies $\mathbb{P}_{\hat{\theta}, x \sim \mathcal{D}}\left(f_{\hat{\theta}}(x) \in (1 \pm \varepsilon)f_\theta(x)\right) \geq 1 - \delta$, and the number of filters in $\hat{\theta}$ is bounded by $\mathcal{O}\left(\sum_{\ell=1}^{L} \frac{L^2 (\Delta^{\ell \rightarrow})^2 S^\ell \log(\eta/\delta)}{\varepsilon^2}\right)$.*

## 3 Relative Layer Importance

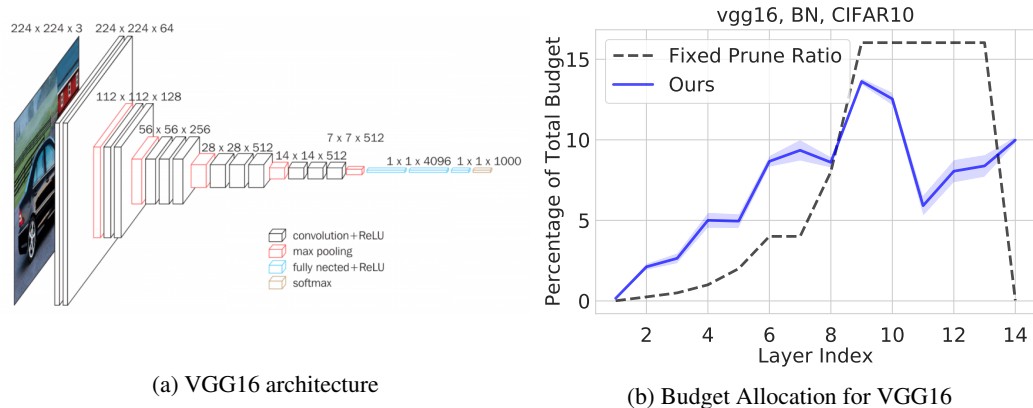

(a) VGG16 architecture

(b) Budget Allocation for VGG16

Figure 2: Early layers of VGG are relatively harder to approximate due to their large spatial dimensions as shown in (a). Our error bounds naturally bridge layer compressibility and importance and enable us to automatically allocate relatively more samples to early layers and less to latter layers as shown in (b). The final layer – due to its immediate influence on the output – is also automatically assigned a large portion of the sampling budget.

In the previous sections, we established the sampling complexity of our filter pruning scheme for any user-specified $\varepsilon$ and $\delta$. However, in practice, it is more common for the practitioner to specify the desired pruning ratio, which specifies the resulting size of the pruned model. Given this sampling budget, a practical question that arises is how to optimally ration the sampling budget across the network's layers to minimize the error of the pruned model. A naive approach would be to uniformly allocate the sampling budget $N$ so that the same ratio of filters is kept in each layer. However, this allocation scheme implicitly assumes that each layer of the network is of equal importance to retaining the output, which is virtually never the case in practice, as exemplified by Fig. 2(a).

It turns out that our analytical bounds on the sample complexity per layer ($m^\ell$ in Alg. 1) naturally capture the importance of each layer. The key insight lies in bridging the compressibility and

importance of each layer: if a layer is not very important, i.e., it does not heavily influence output of the network, then we expect it to be highly compressible, and vice-versa. This intuition is precisely captured by our sampling complexity bounds that quantify the difficulty of a layer's compressibility.

We leverage this insight to formulate a simple binary search procedure for judiciously allocating the sampling budget $N$ as follows. Let $\delta \in (0, 1)$ be user-specified, pick a random $\varepsilon > 0$, and compute the sampling complexity $m^\ell$ as in Alg. 1 together with the resulting layer size $n^\ell$. If $\sum_\ell n^\ell = N$, we are done, otherwise, continue searching for an appropriate $\varepsilon$ on a smaller interval depending on whether $\sum_\ell n^\ell$ is greater or less than $N$. The allocation generated by this procedure (see Fig. 2(b) for an example) ensures that the maximum layer-wise error incurred by pruning is at most $\varepsilon$.

## 4 Results

In this section, we evaluate and compare our algorithm's performance to that of state-of-the-art pruning schemes in generating compact networks that retain the predictive accuracy of the original model. Our evaluations show that our approach generates significantly smaller and more efficient models compared to those generated by competing methods. Our results demonstrate the practicality and wide-spread applicability of our proposed approach: across all of our experiments, our algorithm took on the order of a minute to prune a given network[5], required no manual tuning of its hyper-parameters, and performed consistently well across a diverse set of pruning scenarios. Additional results, comparisons, and experimental details can be found in Sec. E of the appendix.

### 4.1 Experimental Setup

We compare our algorithm to that of the following filter pruning algorithms that we implemented and ran alongside our algorithm: Filter Thresholding (FT, Li et al. (2016)), SoftNet (He et al., 2018), and ThiNet (Luo et al., 2017). We note that FT and SoftNet are both (weight) magnitude-based filter pruning algorithms, and this class of pruning schemes has recently been reported to be state-of-the-art (Gale et al., 2019; Pitas et al., 2019; Yu et al., 2018) (see Sec. E.1 of the appendix for details of the compared methods). Additional comparisons to other state-of-the-art channel and filter pruning methods can be found in Tables 6 and 8 in Appendix E.4 and E.6, respectively.

Our algorithm only requires two inputs in practice: the desired pruning ratio (PR) and failure probability $\delta \in (0, 1)$, since the number of samples in each layer is automatically assigned by our allocation procedure described in Sec. 3. Following the conventional data partioning ratio, we reserve 90% of the training data set for training and the remaining 10% for the validation set (Lee et al., 2019).

For each scenario, we prune the original (pre-trained) network with a target prune ratio using the respective pruning algorithm and fine-tune the network by retraining for a specified number of epochs. We repeat this procedure iteratively to obtain various target prune ratios and report the percentage of parameters pruned (PR) and the percent reduction in FLOPS (FR) for each target prune ratio. The target prune ratio follows a hyperharmonic sequence where the $i^{\text{th}}$ PR is determined by $1 - 1/(i+1)^\alpha$, where $\alpha$ is an experiment-dependent tuning parameter. We conduct the prune-retrain cycle for a range of $10 - 20$ target prune ratios, and report the highest PR and FR for which the compressed network achieves commensurate accuracy, i.e., when the pruned model's test accuracy is within 0.5% of the original model. The quantities reported are averaged over 3 trained models for each scenario, unless stated otherwise. The full details of our experimental setup and the hyper-parameters used can be found in the appendix (Sec. E).

### 4.2 LeNet architectures on MNIST

As our first experiment, we evaluated the performance of our pruning algorithm and the comparison methods on LeNet300-100 (LeCun et al., 1998), a fully-connected network with two hidden layers of size 300 and 100 hidden units, respectively, and its convolutional counterpart, LeNet-5 (LeCun et al., 1998), which consists of two convolutional layers and two fully-connected layers. Both networks were trained on MNIST using the hyper-parameters specified in Sec. E.

---

[5]Excluding the time required for the fine-tuning step, which was approximately the same across all methods

Table 1 depicts the performance of each pruning algorithm in attaining the sparsest possible network that achieves commensurate accuracy for the LeNet architectures. In both scenarios, our algorithm generates significantly sparser networks compared to those generated by the competing filter pruning approaches. In fact, the pruned LeNet-5 model generated by our algorithm by removing filters achieves a prune ratio of $\approx 90\%$, which is even competitive with the accuracy of the sparse models generated by state-of-the-art *weight pruning* algorithms (Lee et al., 2019) [6]. In addition to evaluating the sparsity of the generated models subject to the commensurate accuracy constraint, we also investigated the performance of the pruning algorithms for extreme (i.e., around $5\%$) pruning ratios (see Fig. 3(a)). We see that our algorithm's performance relative to those of competing algorithms is strictly better for a wide range of target prune ratios. For LeNet-5 Fig. 3(a) shows that our algorithm's favorable performance is even more pronounced at extreme sparsity levels (at $\approx 95\%$ prune ratio).

| [%] | Method | Err. | PR |
|---|---|---|---|
| LeNet-300-100 | Unpruned | 1.59 | |
| | Ours | +0.41 | **84.32** |
| | FT | +0.35 | 81.68 |
| | SoftNet | +0.41 | 81.69 |
| | ThiNet | +10.58 | 75.01 |
| LeNet-5 | Unpruned | 0.72 | |
| | Ours | +0.35 | **92.37** |
| | FT | +0.47 | 85.04 |
| | SoftNet | +0.40 | 80.57 |
| | ThiNet | +0.12 | 58.17 |

Table 1: The prune ratio (PR) and the corresponding test error (Err.) of the sparsest network – with commensurate accuracy – generated by each algorithm.

## 4.3 Convolutional Neural Networks on CIFAR-10

Next, we evaluated the performance of each pruning algorithm on significantly larger and deeper Convolutional Neural Networks trained on the CIFAR-10 data set: VGG16 with BatchNorm (Simonyan & Zisserman, 2015), ResNet20, ResNet56, ResNet110 (He et al., 2016), DenseNet22 (Huang et al., 2017), and WideResNet16-8 (Zagoruyko & Komodakis, 2016). For CIFAR-10 experiments, we use the standard data augmentation techniques: padding 4 pixels on each side, random crop to 32x32 pixels, and random horizontal flip. Our results are summarized in Table 2 and Figure 3. Similar to the results reported in Table 1 in the previous subsection, Table 2 shows that our method is able to achieve the most sparse model with minimal loss in predictive power relative to the original network. Furthermore, by inspecting the values reported for ratio of Flops pruned (FR), we observe that the models generated by our approach are not only more sparse in terms of the number of total parameters, but also more efficient in terms of the inference time complexity.

| [%] | Orig. Err. | Ours | | | FT | | | SoftNet | | | ThiNet | | |
|---|---|---|---|---|---|---|---|---|---|---|---|---|---|
| | | Err. | PR | FR | Err. | PR | FR | Err. | PR | FR | Err. | PR | FR |
| ResNet20 | 8.60 | +0.49 | **62.67** | 45.46 | +0.43 | 42.65 | 44.59 | +0.50 | 46.42 | **49.40** | +2.10 | 32.90 | 32.73 |
| ResNet56 | 7.05 | +0.28 | **88.98** | 84.42 | +0.48 | 81.46 | 82.73 | +0.36 | 81.46 | 82.73 | +1.28 | 50.08 | 50.06 |
| ResNet110 | 6.43 | +0.36 | **92.07** | 89.76 | +0.17 | 86.38 | 87.39 | +0.34 | 86.38 | 87.39 | +0.92 | 49.70 | 50.39 |
| VGG16 | 7.11 | +0.50 | **94.32** | 85.03 | +1.11 | 80.09 | 80.14 | +0.81 | 63.95 | 63.91 | +2.13 | 63.95 | 64.02 |
| DenseNet22 | 10.07 | +0.46 | **56.44** | 62.66 | +0.32 | 29.31 | 30.23 | +0.21 | 29.31 | 30.23 | +4.36 | 50.76 | 51.06 |
| WRN16-8 | 4.83 | +0.46 | **66.22** | 64.57 | +0.40 | 24.88 | 24.74 | +0.14 | 16.93 | 16.77 | +0.35 | 14.18 | 14.09 |

Table 2: Overview of the pruning performance of each algorithm for various CNN architectures. For each algorithm and network architecture, the table reports the prune ratio (PR, %) and pruned Flops ratio (FR, %) of pruned models when achieving test accuracy within 0.5% of the original network's test accuracy (or the closest result when the desired test accuracy was not achieved for the range of tested PRs). Our results indicate that our pruning algorithm generates smaller and more efficient networks with minimal loss in accuracy, when compared to competing approaches.

Fig. 3 depicts the performance of the evaluated algorithms for various levels of prune ratios. Once again, we see the consistently better performance of our algorithm in generating sparser models that approximately match or exceed the predictive accuracy of the original uncompressed network. In addition, Table 8 (see Appendix E.6 for more details) provides further comparisons to state-of-the-art

---

[6] Weight pruning approaches can generate significantly sparser models with commensurate accuracy than can filter pruning approaches since the set of feasible solutions to the problem of filter pruning is a subset of the feasible set for the weight pruning problem

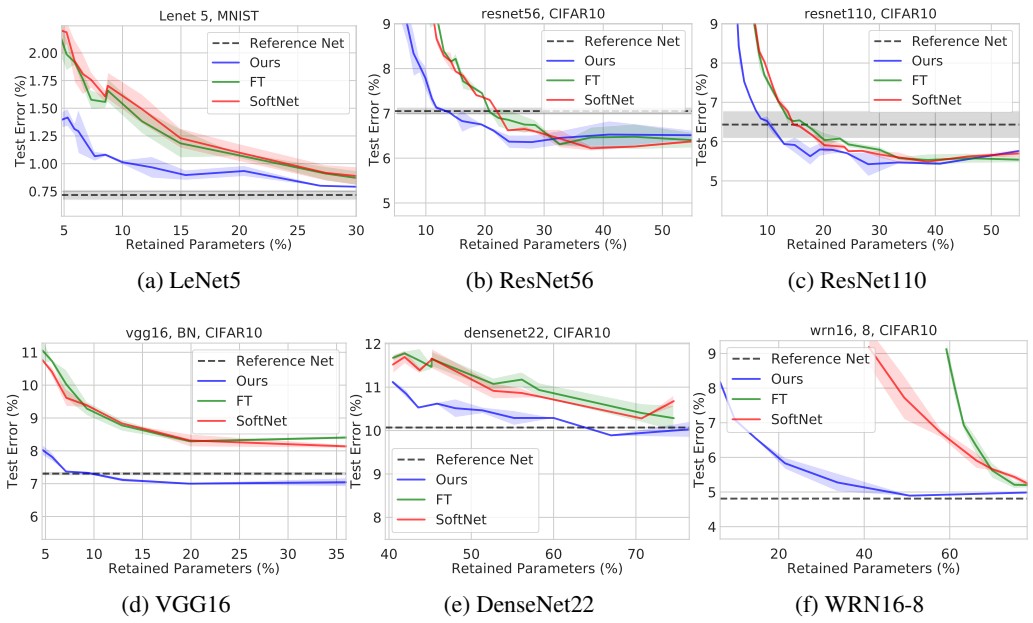

(a) LeNet5        (b) ResNet56        (c) ResNet110

(d) VGG16        (e) DenseNet22       (f) WRN16-8

Figure 3: The accuracy of the generated pruned models for the evaluated pruning schemes for various target prune ratios. Note that the $x$ axis is the percentage of **parameters retained**, i.e., $(1 - \mathrm{pruneratio})$. ThiNet was omitted from the plots for better readability. Our results show that our approach generates pruned networks with minimal loss in accuracy even for high prune ratios. Shaded regions correspond to values within one standard deviation of the mean.

filter pruning methods where we compare the performance of our approach to the results for various ResNets and VGG16 reported directly in the respective papers. The comparisons in Table 8 reaffirm that our algorithm can consistently generate simultaneously sparser and more accurate networks compared to competing methods.

In view of our results from the previous subsection, the results shown in Table 2, Fig. 3, and Table 8 highlight the versatility and broad applicability of our method, and seem to suggest that our approach fares better relative to the compared algorithms on more challenging pruning tasks that involve large-scale networks. We suspect that these favorable properties are explained by the data-informed evaluations of filter importance and the corresponding theoretical guarantees of our algorithm – which enable robustness to variations in network architecture and data distribution.

### 4.4 Convolutional Neural Networks on ImageNet

We consider pruning convolutional neural networks of varying size – ResNet18, ResNet50, and ResNet101 – trained on the ImageNet (Russakovsky et al., 2015) data set. For this dataset, we considered two scenarios: (i) iterative pruning without retraining and (ii) iterative prune-retrain with a limited amount of iterations given the resource-intensive nature of the experiments. The results of these experiments are reported in Section E.4 of the appendix. Our results on the ImageNet data set follow a similar trend as those in the previous subsections and indicate that our method readily scales to larger data sets without the need of manual hyperparameter tuning. This improves upon existing approaches (such as those in He et al. (2018); Li et al. (2016)) that generally require tedious, task-specific intervention or manual parameter tuning by the practitioner.

### 4.5 Application to Real-time Regression Tasks

Real-time applications of neural networks, such as their use in autonomous driving scenarios, require network models that are not only highly accurate, but also highly efficient, i.e., fast, when it comes to inference time complexity (Amini et al., 2018). Model compression, and in particular, filter pruning has potential to generate compressed networks capable of achieving both of these objectives. To evaluate and compare the effectiveness of our method on pruning networks intended for regression

tasks and real-time systems, we evaluated the various pruning approaches on the *DeepKnight* network (Amini et al., 2018), a regression network deployed on an autonomous vehicle in real time to predict the steering angle of the human driver (see E.5 in appendix for experimental details).

Fig. 4 depicts the results of our evaluations and comparisons on the *DeepKnight* network *without* the fine-tuning step. We omitted the iterative fine-tuning step for this scenario and instead evaluated the test loss for various prune ratios because (i) the evaluated algorithms were able to generate highly accurate models without the retraining step and (ii) in order to evaluate and compare the performance of solely the core pruning procedure. Similar to the results obtained in the preceding pruning scenarios, Fig. 4 shows that our method consistently outperforms competing approaches for all of the specified prune ratios.

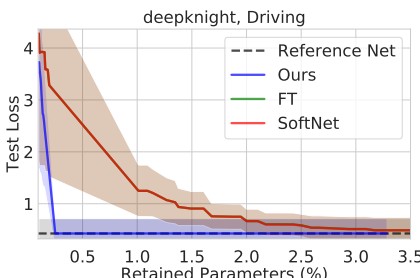

Figure 4: The performance of the compared algorithms on pruning a lightweight network for a real-time regression task (Amini et al., 2018).

### 4.6 DISCUSSION

In addition to the favorable empirical results of our algorithm, our approach exhibits various advantages over competing methods that manifest themselves in our empirical evaluations. For one, our algorithm does not require any additional hyper-parameters other than the pruning ratio and the desired failure probability. Given these sole two parameters, our approach automatically allocates the number of filters to sample for each layer. This alleviates the need to perform time-intensive ablation studies (He et al., 2018) and to resort to uninformed (i.e., uniform) sample allocation strategies, e.g., removing the same percentage of filters in each layer (Li et al., 2016), which fails to consider the non-uniform influence of each layer on the network's output (see Sec. 3). Moreover, our algorithm is simple-to-implement and computationally efficient both in theory and practice: the computational complexity is dominated by the $|\mathcal{S}|$ forward passes required to compute the sensitivities ($|\mathcal{S}| \leq 256$ in practical settings) and in practice, our algorithm takes on the order of a minute to prune the network.

## 5 CONCLUSION

We presented – to the best of our knowledge – the first filter pruning algorithm that generates a pruned network with theoretical guarantees on the size and performance of the generated network. Our method is data-informed, simple-to-implement, and efficient both in theory and practice. Our approach can also be broadly applied to varying network architectures and data sets with minimal hyper-parameter tuning necessary. This stands in contrast to existing filter pruning approaches that are generally data-oblivious, rely on heuristics for evaluating the parameter importance, or require tedious hyper-parameter tuning. Our empirical evaluations on popular network architectures and data sets reaffirm the favorable theoretical properties of our method and demonstrate its practical effectiveness in obtaining sparse, efficient networks. We envision that besides its immediate use for pruning state-of-the-art models, our approach can also be used as a sub-procedure in other deep learning applications, e.g., for identifying winning lottery tickets (Frankle & Carbin, 2019) and for efficient architecture search (Liu et al., 2019b).

ACKNOWLEDGMENTS

This research was supported in part by the U.S. National Science Foundation (NSF) under Awards 1723943 and 1526815, Office of Naval Research (ONR) Grant N00014-18-1-2830, Microsoft, and JP Morgan Chase.

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

# A  RELATED WORK

**General network compression**  The need to tame the excessive storage requirements and costly inference associated with large, over-parameterized networks has led to a rich body of work in network pruning and compression. These approaches range from those inspired by classical tensor decompositions (Yu et al., 2017b; Jaderberg et al., 2014; Denton et al., 2014), and random projections and hashing (Arora et al., 2018; Ullrich et al., 2017; Chen et al., 2015; Weinberger et al., 2009; Shi et al., 2009) that compress a pre-trained network, to those approaches that enable sparsity by embedding sparsity as an objective directly in the training process (Ioannou et al., 2015; Alvarez & Salzmann, 2017) or exploit tensor structure to induce sparsity (Choromanska et al., 2016; Zhao et al., 2017). Overall, the predominant drawback of these methods is that they require laborious hyperparameter tuning, lack rigorous theoretical guarantees on the size and performance of the resulting compressed network, and/or conduct compression in a data oblivious way.

**Weight-based pruning**  A large subset of modern pruning algorithms fall under the general approach of pruning individual weights of the network by assigning each weight a saliency score, e.g., its magnitude (Han et al., 2015), and subsequently inducing sparsity by deterministically removing those weights below a certain saliency score threshold (Guo et al., 2016; Han et al., 2015; Lee et al., 2019; LeCun et al., 1990). These approaches are heuristics that do not provide any theoretical performance guarantees and generally require – with the exception of (Lee et al., 2019) – computationally expensive train-prune-retrain cycles and tedious hyper-parameter tuning. Unlike our approach that enables accelerated inference (i.e., reduction in FLOPS) on any hardware and with any deep learning library by generating a smaller subnetwork, weight-based pruning generates a model with non-structured sparsity that requires specialized hardware and sparse linear algebra libraries in order to speed up inference.

**Neuron pruning**  Pruning entire neurons in FNNs and filters in CNNs is particularly appealing as it shrinks the network into its slimmer counterpart, which leads to alleviated storage requirements and improved inference-time performance on any hardware. Similar to the weight-based approaches, approaches in this domain assign an importance score to each neuron or filter and remove those with a score below a certain threshold (He et al., 2018; Li et al., 2016; Yu et al., 2017a). These approaches generally take the $\ell_p$ norm –with $p = \{1, 2\}$ as popular choices– of the filters to assign filter importance and subsequently prune unimportant filers. These methods are data-oblivious heuristics that heavily rely on the assumption that filters with large weight magnitudes are more important, which may not hold in general (Ye et al., 2018).

In general, prior work on neuron and filter pruning has focused on approaches that lack theoretical guarantees and a principled approach to allocating the sampling budget across layers, requiring tedious ablation studies or settling for naive uniform allocation across the layers. In contrast to prior approaches, our algorithm assigns data-informed saliency scores to filters, guarantees an error bound, and leverages our theoretical error bounds to automatically identify important layers and allocate the user-specified sampling budget (i.e., pruning ratio) across the layers.

Our work is most similar to that of (Baykal et al., 2019a;b), which proposed an weight pruning algorithm with provable guarantees that samples weights of the network in accordance to an empirical notion of parameter importance. The main drawback of their approach is the limited applicability to only fully-connected networks, and the lack of inference-time acceleration due to non-structured sparsity caused by removing individual weights. Our method is also sampling-based and relies on a data-informed notion of importance, however, unlike (Baykal et al., 2019a;b), our approach can be applied to both FNNs and CNNs and generates sparse, efficient subnetworks that accelerate inference.

# B  ALGORITHMIC AND ANALYTICAL DETAILS

Algorithm 2 is the full algorithm for pruning features, i.e., neurons in fully-connected layers and channels in convolutional layers. For notational simplicity, we will derive our theoretical results for linear layers, i.e., neuron pruning. We remind the reader that this result also applies to CNNs by taking channels of a weight tensor in place of neurons. The pseudocode is organized for clarity of exposition rather than computational efficiency. Recall that $\theta$ is the full parameter set of the net,

where $W^\ell \in \mathbb{R}^{\eta^\ell \times \eta^{\ell+1}}$ is the weight matrix between layers $\ell - 1$ and and $\ell$. $W_k^\ell$ refers to the $k^{\text{th}}$ neuron of $W^\ell$.

---

**Algorithm 2** PRUNECHANNELS($\theta, \ell, \mathcal{S}, \varepsilon, \delta$) - *extended version*

---

**Input:** $\theta$: trained net; $\ell \in [L]$: layer; $\mathcal{S} \subset \text{supp}(\mathcal{D})$: sample of inputs; $\varepsilon \in (0, 1)$: accuracy; $\delta \in (0, 1)$: failure probability
**Output:** $\hat{W}^\ell$: filter-reduced weight tensor for layer $\ell$; $\hat{W}^{\ell+1}$: channel reduced, weight tensor for layer $\ell + 1$

1: **for** $j \in [\eta^\ell]$ **do**
2:     **for** $i \in [\eta^{\ell+1}]$ and $\mathbf{x} \in \mathcal{S}$ **do**
3:         $I^+ \leftarrow \{j \in [\eta^\ell] : w_{ij}^{\ell+1} a_j^\ell(\mathbf{x}) \geq 0\}$
4:         $I^- \leftarrow [\eta^\ell] \setminus I^+$
5:         $g_{ij}^{\ell+1}(\mathbf{x}) \leftarrow \max_{I \in \{I^+, I^-\}} \frac{w_{ij}^{\ell+1} a_j^\ell(\mathbf{x})}{\sum_{k \in I} w_{ik}^{\ell+1} a_k^\ell(\mathbf{x})}$
6:     **end for**
7:     $s_j^\ell \leftarrow \max_{\mathbf{x} \in \mathcal{S}} \max_{i \in [\eta^{\ell+1}]} g_{ij}^{\ell+1}(\mathbf{x})$
8: **end for**
9: $S^\ell \leftarrow \sum_{j \in [\eta^\ell]} s_j^\ell$
10: **for** $j \in [\eta^\ell]$ **do**
11:     $p_j^\ell \leftarrow s_j^\ell / S^\ell$
12: **end for**
13: $K \leftarrow$ value from Assumption 1
14: $m \leftarrow \lceil (6 + 2\varepsilon) S^\ell K \log(2\eta^{\ell+1}/\delta)\varepsilon^{-2} \rceil$
15: $\mathcal{H} \leftarrow$ distribution on $[\eta^\ell]$ assigning probability $p_j^\ell$ to index $j$
16: $\hat{W}^\ell \leftarrow (0, \ldots, 0)$ {same dimensions as $W^\ell$}
17: $\hat{W}^{\ell+1} \leftarrow (0, \ldots, 0)$ {same dimensions as $W^{\ell+1}$}
18: **for** $k \in [m]$ **do**
19:     $c(k) \leftarrow$ random draw from $\mathcal{H}$
20:     $\hat{W}_{c(k)}^\ell \leftarrow W_{c(k)}^\ell$ {no reweighing or considering multiplicity of drawing index $c(k)$ multiple times}
21:     $\hat{W}_{:c(k)}^{\ell+1} \leftarrow \hat{W}_{:c(k)}^{\ell+1} + \frac{W_{:c(k)}^{\ell+1}}{m p_{c(k)}}$ {reweighing for unbiasedness of pre-activation in layer $\ell + 1$}
22: **end for**
23: **return** $\hat{W}^\ell = [\hat{W}_1^\ell, \ldots, \hat{W}_{\eta^\ell}^\ell]; \hat{W}^{\ell+1} = [\hat{W}_{:1}^{\ell+1}, \ldots, \hat{W}_{:\eta^\ell}^{\ell+1}]$

---

Recall that $z_i^{\ell+1}(\mathbf{x})$ denotes the pre-activation of the $i^{\text{th}}$ neuron in layer $\ell + 1$ given input $\mathbf{x}$, and the activation $a_j^\ell(x) = \max\{0, z_j^\ell(\mathbf{x})\}$.

**Definition 1** (Edge Sensitivity (Baykal et al., 2019a)). *Fixing a layer $\ell \in [L]$, let $w_{ij}^{\ell+1}$ be the weight of edge $(j, i) \in [\eta^\ell] \times [\eta^{\ell+1}]$. The empirical sensitivity of weight entry $w_{ij}^{\ell+1}$ with respect to input $\mathbf{x} \in \mathcal{X}$ is defined to be*

$$g_{ij}^{\ell+1}(\mathbf{x}) = \max_{I \in \{I^+, I^-\}} \frac{w_{ij}^{\ell+1} a_j^\ell(\mathbf{x})}{\sum_{k \in I} w_{ik}^{\ell+1} a_k^\ell(\mathbf{x})}, \tag{2}$$

*where $I^+ = \{j \in [\eta^\ell] : w_{ij}^{\ell+1} a_j^\ell(\mathbf{x}) \geq 0\}$ and $I^- = [\eta^\ell] \setminus I^+$ denote the set of positive and negative edges, respectively.*

Algorithm 2 uses empirical sensitivity to compute the sensitivity of neurons on Lines 9-12.

**Definition 2** (Neuron Sensitivity). *The sensitivity of a neuron $j \in [\eta^\ell]$ in layer $\ell$ is defined as*

$$s_j^\ell = \max_{\mathbf{x} \in \mathcal{S}} \max_{i \in [\eta^{\ell+1}]} g_{ij}^{\ell+1}(\mathbf{x}) \tag{3}$$

In this section, we prove that Algorithm 2 yields a good approximation of the original net. We begin with a mild assumption to ensure that the distribution of our input is not pathological.

**Assumption 1.** *There exist universal constants $K, K' > 0$ such that for any layer $\ell$ and all $j \in [\eta^\ell]$, the CDF of the random variable $\max_{i \in [\eta^{\ell+1}]} g_{ij}^{\ell+1}(x)$ for $x \sim \mathcal{D}$, denoted by $F_j(\cdot)$, satisfies*

$$F_j(M_j/K) \leq \exp(-1/K'),$$

*where $M_j = \min\{y \in [0, 1] : F_j(y) = 1\}$.*

Note that the analysis is carried out for the positive and negative elements of $W^{\ell+1}$ separately, which is also considered in the definition of sensitivity (Def. 1). For ease of exposition, we will thus assume that throughout the section $W^{\ell+1} \geq 0$ (element-wise), i.e., $I^+ = [\eta^\ell]$, and derive the results for this case. However, we note that we could equivalently assume $W^{\ell+1} \leq 0$ and the analysis would hold regardless. By considering both the positive and negative parts of $W^{\ell+1}$ in Def. 1 we can carry out the analysis for weight tensors with positive and negative elements.

**Theorem 2.** *Let $\varepsilon, \delta \in (0,1), \ell \in [L]$, and let $\mathcal{S}$ be a set of $\Theta(\log(\eta_*/\delta))$ i.i.d. samples drawn from $\mathcal{D}$. Then, $\hat{W}^{\ell+1}$ contains at most $\mathcal{O}(S^\ell \log(\eta_*/\delta)\varepsilon^{-2})$ channels and for $x \sim \mathcal{D}$, with probability at least $1 - \delta$, we have $\hat{z}^{\ell+1} \in (1 \pm \varepsilon)z^{\ell+1}$ (entry-wise), where $\eta_* = \max_{\ell \in [L]} \eta^\ell$.*

The remainder of this section builds towards proving Theorem 2. We begin by fixing a layer $\ell \in [L]$ and neuron $i \in [\eta^{\ell+1}]$. Consider the random variables $\{Y_k\}_{k \in [m]}$ where $Y_k(\mathbf{x}) = \frac{1}{mp_j}w_{ij}^{\ell+1}a_j^\ell(\mathbf{x})$ where Algorithm 2 selected index $j \in [\eta^\ell]$ on the $k^{\text{th}}$ iteration of Line 19. Note that $z_i^{\ell+1}(\mathbf{x}) = \sum_{j \in [\eta^\ell]} w_{ij}^{\ell+1}a_j^\ell(\mathbf{x})$ and so we may also write $g_{ij}^{\ell+1}(\mathbf{x}) = w_{ij}^{\ell+1}a_j^\ell(\mathbf{x})/z_i^{\ell+1}(\mathbf{x})$ when it is more convenient.

**Lemma 4.** *For each $\mathbf{x} \in \mathcal{X}$ and $k \in [m]$, $\mathbb{E}[Y_k(\mathbf{x})] = z_i^{\ell+1}(\mathbf{x})/m$.*

*Proof.* $Y_j$ is drawn from distribution $\mathcal{H}$ defined on Line 15, so we compute the expectation directly.

$$
\begin{aligned}
\mathbb{E}[Y_j(\mathbf{x})] &= \sum_{k \in [\eta^\ell]} \frac{w_{ik}^{\ell+1}a_k^\ell(\mathbf{x})}{m\,p_k} \cdot p_k \\
&= \frac{1}{m} \sum_{k \in [\eta^\ell]} w_{ik}^{\ell+1}a_k^\ell(\mathbf{x}) \\
&= \frac{z_i^{\ell+1}(\mathbf{x})}{m}
\end{aligned}
$$

$\square$

To bound the variance, we use an approach inspired by Baykal et al. (2019a) where the main idea is to use the notion of empirical sensitivity to establish that a particular useful inequality holds with high probability over the randomness of the input point $x \sim \mathcal{D}$. Given that the inequality holds we can establish favorable bounds on the variance and magnitude of the random variables, which lead to a low sampling complexity.

For a random input point $\mathbf{x} \sim \mathcal{D}$, let $\mathcal{G}$ denote the event that the following inequality holds (for all neurons):

$$
\max_{i \in [\eta^{\ell+1}]} g_{ij}^{\ell+1}(\mathbf{x}) \leq C\,s_j \quad \forall j \in [\eta^\ell]
$$

where $C = \max\{3K, 1\}$ and $K$ is defined as in Assumption 1. We now prove that under Assumption 1, event $\mathcal{G}$ occurs with high probability. From now on, to ease notation, we will drop certain superscripts/subscripts with the meaning is clear. For example, $z(\mathbf{x})$ will refer to $z_i^{\ell+1}(\mathbf{x})$.

**Lemma 5.** *If Assumption 1 holds, $\mathbb{P}(\mathcal{G}) > 1 - \delta/2\eta^\ell$. Here the probability is over the randomness of drawing $\mathbf{x} \sim \mathcal{D}$.*

*Proof.* Since $\max_{i \in [\eta^{\ell+1}]} g_{ij}(x)$ is just a function of the random variable $x \sim \mathcal{D}$, for any $j \in [\eta^\ell]$ we can let $D$ be a distribution over $\max_{i \in [\eta^{\ell+1}]} g_{ij}(x)$ and observe that since $s_j = \max_{\mathbf{x} \in \mathcal{S}} \max_{i \in [\eta^{\ell+1}]} g_{ij}(\mathbf{x})$, the negation of event $\mathcal{G}$ for a single neuron $j \in [\eta^\ell]$ can be expressed as the event

$$
X > C \max_{k \in [|\mathcal{S}|]} X_k,
$$

where $X \sim D$ and $X_1, \ldots, X_{|\mathcal{S}|} \overset{i.i.d.}{\sim} D$ since the points in $\mathcal{S}$ were drawn i.i.d. from $\mathcal{D}$. Invoking Lemma 8 from Baykal et al. (2019a) in conjunction with Assumption 1, we obtain for any arbitrary $j$

$$
\mathbb{P}(\max_{i \in [\eta^{\ell+1}]} g_{ij}(x) > C\,s_j) = \mathbb{P}(X > C \max_{k \in [|\mathcal{S}|]} X_k) \leq \exp(-|\mathcal{S}|/K')
$$

with the $K'$ from Assumption 1. Since our choice of neuron $j$ was arbitrary, the inequality above holds for all neurons, therefore we can apply the union bound to obtain:

$$
\begin{aligned}
\mathop{\mathbb{P}}_{x \sim \mathcal{D}}(\mathcal{G}) &= 1 - \mathbb{P}(\exists j \in [\eta^\ell] : \max_{i \in [\eta^{\ell+1}]} g_{ij}(x) > C\, s_j) \\
&\geq 1 - \sum_{j \in [\eta^\ell]} \mathbb{P}(\max_{i \in [\eta^{\ell+1}]} g_{ij}(x) > C\, s_j) \\
&\geq 1 - \eta^\ell \, \exp(-|\mathcal{S}|/K') \\
&\geq 1 - \frac{\delta}{2\eta^{\ell+1}}
\end{aligned}
$$

where the last line follows from the fact that $|\mathcal{S}| \geq \lceil K' \log\left(2\,\eta^\ell\eta^{\ell+1}/\delta\right) \rceil$. $\qquad \square$

**Lemma 6.** *For any* $\mathbf{x}$ *such that event* $\mathcal{G}$ *occurs, then* $|Y_k(\mathbf{x}) - \mathbb{E}\left[Y_k(\mathbf{x})\right]| \leq CSz/m$. *Here the expectation is over the randomness of Algorithm 2.*

*Proof.* Recall that $S = \sum_{j \in [\eta^\ell]} s_j$. Let neuron $j \in [\eta^\ell]$ be selected on iteration $k$ of Line 19. For any $k \in [m]$ we have:

$$
\begin{aligned}
Y_k(\mathbf{x}) &= \frac{w_{ij} a_j(\mathbf{x})}{m p_j} \\
&= S \frac{w_{ij} a_j(\mathbf{x})}{m\, s_j} \\
&\leq C S \frac{w_{ij} a_j(\mathbf{x})}{m\, \max_{i'} g_{i'j}(\mathbf{x})} \\
&\leq C S \frac{w_{ij} a_j(\mathbf{x})}{m\, g_{ij}(\mathbf{x})} \\
&= \frac{C S z}{m},
\end{aligned}
$$

where the first inequality follows by the inequality of event $\mathcal{G}$, the second by the fact that $\max_{i'} g_{i'j}(\mathbf{x}) \geq g_{ij}(\mathbf{x})$ for any $i$, and the third equality by definition of $g_{ij}(\mathbf{x}) = w_{ij} a_j(\mathbf{x})/z(\mathbf{x})$. This implies that $|Y_k - \mathbb{E}\left[Y_k\right]| = \left|Y_k - \frac{z}{m}\right| \in [-z/m, CSz/m]$ by Lemma 4 and since $Y_k \geq 0$. The result follows since $C, S \geq 1$. $\qquad \square$

**Lemma 7.** *For any* $\mathbf{x}$ *such that event* $\mathcal{G}$ *occurs, then* $\mathrm{Var}(Y_k(\mathbf{x})) \leq CSz^2/m^2$. *Here the expectation is over the randomness of Algorithm 2.*

*Proof.* We can use the same inequality obtained by conditioning on $\mathcal{G}$ to bound the variance of our estimator.

$$
\begin{aligned}
\mathrm{Var}(Y_k(\mathbf{x})) &= \mathbb{E}\left[Y_k^2(\mathbf{x})\right] - \left(\mathbb{E}\left[Y_k(\mathbf{x})\right]\right)^2 \\
&\leq \mathbb{E}\left[Y_k^2(\mathbf{x})\right] \\
&= \sum_{j \in [\eta^\ell]} \left(\frac{w_{ij} a_j(\mathbf{x})}{m\, p_j}\right)^2 \cdot p_j && \text{by definition of } Y_k \\
&= \frac{S}{m^2} \sum_{j \in [\eta^\ell]} \frac{(w_{ij} a_j(\mathbf{x}))^2}{s_j} && \text{since } p_j = s_j/S \\
&\leq \frac{CS}{m^2} \sum_{j \in [\eta^\ell]} \frac{(w_{ij} a_j(\mathbf{x}))^2}{\max_{i' \in [\eta^{\ell+1}]} g_{i'j}(\mathbf{x})} && \text{by occurrence of event } \mathcal{G} \\
&\leq \frac{CS z}{m^2} \sum_{j \in [\eta^\ell]} w_{ij} a_j(\mathbf{x}) && \text{since } g_{ij}(\mathbf{x}) = w_{ij} a_j(\mathbf{x})/z(\mathbf{x}) \\
&= \frac{CSz^2}{m^2}.
\end{aligned}
$$

$\square$

We are now ready to prove Theorem 2.

*Proof of Theorem 2.* Recall the form of Bernstein's inequality that, given random variables $X_1, \ldots, X_m$ such that for each $k \in [m]$ we have $\mathbb{E}[X_k] = 0$ and $|X_k| \leq M$ almost surely, then

$$\mathbb{P}\left(\sum_{k \in [m]} X_k \geq t\right) \leq \exp\left(\frac{-t^2/2}{\sum_{k \in [m]} \mathbb{E}[X_k^2] + Mt/3}\right)$$

We apply this with $X_k = Y_k - \frac{z}{m}$. We must take the probability with respect to the randomness of both drawing $\mathbf{x} \sim \mathcal{D}$ and Algorithm 2. By Lemma 4, $E[X_k] = 0$. Let us assume that event $\mathcal{G}$ occurs. By Lemma 6, we may set $M = CSz/m$. By Lemma 7, $\sum_{k \in [m]} \mathbb{E}[X_k^2] \leq CSz^2/m$. We will apply the inequality with $t = \varepsilon z$.

Observe that $\sum_{k \in [m]} X_k = \hat{z} - z$. Plugging in these values, and taking both tails of the inequality, we obtain:

$$\mathbb{P}(|\hat{z} - z| \geq \varepsilon z \, : \, \mathcal{G}) \leq 2\exp\left(\frac{-\varepsilon^2 z^2/2}{CSz^2/m + CS\varepsilon z^2/3m}\right)$$

$$= 2\exp\left(-\frac{\varepsilon^2 m}{SK(6 + 2\varepsilon)}\right) \qquad \text{since } C \leq 3K$$

$$\leq \frac{\delta}{2\eta^{\ell+1}} \qquad \text{by definition of } m$$

Removing dependence on event $\mathcal{G}$, we write:

$$\mathbb{P}(|\hat{z} - z| \geq \varepsilon z) \geq \mathbb{P}(|\hat{z} - z| \geq \varepsilon z \, : \, \mathcal{G})\, \mathbb{P}(\mathcal{G}) \geq \left(1 - \frac{\delta}{2\eta^{\ell+1}}\right)\left(1 - \frac{\delta}{2\eta^{\ell+1}}\right)$$

$$\geq 1 - \frac{\delta}{\eta^{\ell+1}}$$

where we have applied Lemma 5. This implies the result for any single neuron, and the theorem follows by application of the union bound over all $\eta^{\ell+1}$ neurons in layer $\ell$. $\square$

### B.1  BOOSTING SAMPLING VIA DETERMINISTIC CHOICES

Importance sampling schemes, such as the one described above, are powerful tools with numerous applications in Big Data settings, ranging from sparsifying matrices (Baykal et al., 2019a; Achlioptas et al., 2013; Drineas & Zouzias, 2011; Kundu & Drineas, 2014; Tropp et al., 2015) to constructing coresets for machine learning problems (Braverman et al., 2016; Feldman & Langberg, 2011; Bachem et al., 2017). However, by the nature of the exponential decay in probability associated with importance sampling schemes (see Theorem 1), sampling schemes perform truly well when the sampling pool and the number of samples is sufficiently large (Tropp et al., 2015). However, under certain conditions on the sampling distribution, the size of the sampling pool, and the size of the desired sample $m$, it has been observed that deterministically picking the $m$ samples corresponding to the highest $m$ probabilities may yield an estimator that incurs lower error (McCurdy, 2018; Papailiopoulos et al., 2014).

To this end, consider a hybrid scheme that picks $k$ indices deterministically (without reweighing) and samples $m'$ indices. More formally, let $\mathcal{C}_{\det} \subseteq [n]$ be the set of $k$ unique indices (corresponding to weights) that are picked deterministically, and define

$$\hat{z}_{\det} = \sum_{j \in \mathcal{C}_{\det}} w_{ij} a_j,$$

where we note that the weights are not reweighed. Now let $\mathcal{C}_{\mathrm{rand}}$ be a set of $m'$ indices sampled from the remaining indices i.e., sampled from $[n] \setminus \mathcal{C}_{\det}$, with probability distribution $q = (q_1, \ldots, q_n)$.

To define the distribution $q$, recall that the original distribution $p$ is defined to be $p_i = s_i/S$ for each $i \in [n]$. Now, $q$ is simply the normalized distribution resulting from setting the probabilities associated with indices in $\mathcal{C}_{\mathrm{det}}$ to be 0, i.e.,

$$q_i = \begin{cases} \frac{s_i}{S - S_k} & \text{if } i \notin \mathcal{C}_{\mathrm{det}}, \\ 0 & \text{otherwise} \end{cases},$$

where $S_k = \sum_{j \in \mathcal{C}_{\mathrm{det}}} s_j$ is the sum of sensitivities of the entries that were deterministically picked.

Instead of doing a combinatorial search over all $\binom{n}{k}$ choices for the deterministic set $\mathcal{C}_{\mathrm{det}}$, for computational efficiency, we found that setting $\mathcal{C}_{\mathrm{det}}$ to be the indices with the top $k$ sensitivities was the most likely set to satisfy the condition above.

We state the general theorem below.

**Theorem 8.** *It is better to keep $k$ feature maps, $\mathcal{C}_{\mathrm{det}} \subseteq [\eta^\ell]$, $|\mathcal{C}_{\mathrm{det}}| = k$, deterministically and sample $m' = \left\lceil (6 + 2\varepsilon)\, (S^\ell - S_k^\ell)\, K\, \log(8\eta_*/\delta)\varepsilon^{-2} \right\rceil$ features from $[\eta^\ell] \setminus \mathcal{C}_{\mathrm{det}}$ if*

$$\sum_{j \notin \mathcal{C}_{\mathrm{det}}} \left( 1 - \frac{s_j}{S - S_k} \right)^{m'} > \sum_{j=1}^{\eta^\ell} \left( 1 - \frac{s_j}{S} \right)^m + \sqrt{\frac{\log(2/\delta)(m + m')}{2}},$$

*where $m = \left\lceil (6 + 2\varepsilon)\, S^\ell\, K\, \log(4\eta_*/\delta)\varepsilon^{-2} \right\rceil$, $S_k = \sum_{j \in \mathcal{C}_{\mathrm{det}}} s_j$ and $\eta_* = \max_\ell \eta^\ell$.*

*Proof.* Let $m \geq \left\lceil (6 + 2\varepsilon)\, S^\ell\, K\, \log(4\eta_*/\delta)\varepsilon^{-2} \right\rceil$ as in Lemma 2 and note that from Lemma 2, we know that if $\hat{z}$ is our approximation with respect to sampled set of indices, $\mathcal{C}$, we have

$$\mathbb{P}(\mathcal{E}) \leq \delta$$

where $\mathcal{E}$ is the event that the inequality

$$\left| \hat{z}_i^{\ell+1}(x) - z_i^{\ell+1}(x) \right| \leq \varepsilon z_i^{\ell+1}(x) \quad \forall i \in [\eta^{\ell+1}]$$

holds. Henceforth, we will let $i \in [\eta^{\ell+1}]$ be an arbitrary neuron and, similar to before, consider the problem of approximating the neuron's value $z_i^{\ell+1}(x)$ (subsequently denoted by $z$) by our approximating $\hat{z}_i^{\ell+1}(x)$ (subsequently denoted by $\hat{z}$).

Similar to our previous analysis of our importance sampling scheme, we let $\mathcal{C}_{\mathrm{rand}} = \{c_1, \ldots, c_{m'}\}$ denote the multiset of $m'$ neuron indices that are sampled with respect to distribution $q$ and for each $j \in [m']$ define $Y_j = \hat{w}_{ic_j} a_{c_j}$ and let $Y = \sum_{j \in [m']} Y_j$. For clarity of exposition, we define $\hat{z}_{\mathrm{rand}} = Y$ be our approximation with respect to the random sampling procedure, i.e.,

$$\hat{z}_{\mathrm{rand}} = \sum_{j \in \mathcal{C}_{\mathrm{rand}}} \hat{w}_{ij} a_j = Y.$$

Thus, our estimator under this scheme is given by

$$\hat{z}' = \hat{z}_{\mathrm{det}} + \hat{z}_{\mathrm{rand}}$$

Now we want to analyze the sampling complexity of our new estimator $\hat{z}'$ so that

$$\mathbb{P}(|\hat{z}' - z| \geq \varepsilon z) \leq \delta/2.$$

Establishing the sampling complexity for sampling with respect to distribution $q$ is almost identical to the proof of Theorem 2. First, note that $\mathbb{E}\left[ \hat{z}' \mid \mathbf{x} \right] = \hat{z}_{\mathrm{det}} + \mathbb{E}\left[ \hat{z}_{\mathrm{rand}} \mid \mathbf{x} \right]$ since $\hat{z}_{\mathrm{det}}$ is a constant (conditioned on a realization $\mathbf{x}$ of $x \sim \mathcal{D}$). Now note that for any $j \in [m']$

$$\mathbb{E}\left[ Y_j \mid \mathbf{x} \right] = \sum_{k \in [\eta^\ell] \setminus \mathcal{C}_{\mathrm{det}}} \hat{w}_{ik} a_k \cdot q_k$$

$$= \frac{1}{m'} \sum_{k \in [\eta^\ell] \setminus \mathcal{C}_{\mathrm{det}}} w_{ik} a_k$$

$$= \frac{z - \hat{z}_{\text{det}}}{m'},$$

and so $\mathbb{E}\left[\hat{z}_{\text{rand}} \mid \mathbf{x}\right] = \mathbb{E}\left[Y \mid \mathbf{x}\right] = z - \hat{z}_{\text{det}}$.

This implies that $\mathbb{E}\left[\hat{z}'\right] = \hat{z}_{\text{det}} + (z - \hat{z}_{\text{det}}) = z$, and so our estimator remains unbiased. This also yields

$$|Y - \mathbb{E}\left[Y \mid \mathbf{x}\right]| = |\hat{z}_{\text{rand}} - \mathbb{E}\left[\hat{z}_{\text{rand}}\right]| = |\hat{z}_{\text{rand}} + \hat{z}_{\text{det}} - z|$$
$$= |\hat{z}' - z|,$$

which implies that all we have to do to bound the failure probability of the event $|z' - z| \geq \varepsilon z$ is to apply Bernstein's inequality to our estimator $\hat{z}_{\text{rand}} = Y$, just as we had done in the proof of Theorem 2. The only minor change is the variance and magnitude of the random variables $Y_k$ for $k \in [m']$ since the distribution is now with respect to $q$ and not $p$. Proceeding as in the proof of Lemma 6, we have

$$\hat{w}_{ij} a_j(\mathbf{x}) = \frac{w_{ij} a_j(\mathbf{x})}{m' q_j} = (S - S_k) \frac{w_{ij} a_j(\mathbf{x})}{m' s_j}$$
$$\leq \frac{(S - S_k) C z}{m'}.$$

Now, to bound the magnitude of the random variables note that

$$\mathbb{E}\left[Y_j \mid \mathbf{x}\right] = \frac{z - \hat{z}_{\text{det}}}{m'} = \frac{1}{m'} \sum_{j \notin \mathcal{C}_{\text{det}}} w_{ij} a_j \leq \frac{(S - S_k) C z}{m'}.$$

The result above combined with this fact yields for the magnitude of the random variables

$$R' = \max_{j \in [m']} |Y_j - \mathbb{E}\left[Y_j \mid \mathbf{x}\right]| \leq \frac{(S - S_k) C z}{m'},$$

where we observe that the only relative difference to the bound of Lemma 6 is the term $S - S_k$ appears, where $S_k = \sum_{j \in \mathcal{C}_{\text{det}}} s_j$, instead of $S$[7]

Similarly, for the variance of a single $Y_j$

$$\text{Var}(Y_j \mid \mathbf{x}, \mathcal{G}) \leq \sum_{k \in [\eta^\ell] \setminus \mathcal{C}_{\text{det}}} \frac{(w_{ik} a_k(\mathbf{x}))^2}{m'^2 q_k}$$
$$= \frac{S - S_k}{m'^2} \sum_{k \in [\eta^\ell] \setminus \mathcal{C}_{\text{det}}} \frac{(w_{ik} a_k(\mathbf{x}))^2}{s_k}$$
$$\leq \frac{C(S - S_k) z}{m'^2} \sum_{k \in [\eta^\ell] \setminus \mathcal{C}_{\text{det}}} w_{ik} a_k(\mathbf{x})$$
$$\leq \frac{C(S - S_k) z^2 \min\{1, C(S - S_k)\}}{m'^2},$$

where the last inequality follows by the fact that $\sum_{k \in [\eta^\ell] \setminus \mathcal{C}_{\text{det}}} w_{ik} a_k(\mathbf{x}) \leq z$ and by the sensitivity inequality from the proof of Lemma 7

$$\sum_{k \in [\eta^\ell] \setminus \mathcal{C}_{\text{det}}} w_{ik} a_k(\mathbf{x}) \leq C z \sum_{j \in [\eta^\ell] \setminus \mathcal{C}_{\text{det}}} s_j = C z (S - S_k).$$

This implies by Bernstein's inequality and the argument in proof of Theorem 2 that if we sample

$$m' = \left\lceil (6 + 2\varepsilon) \left(S^\ell - S_k^\ell\right) K \log(8\eta_*/\delta)\varepsilon^{-2} \right\rceil$$

times from the distribution $q$, then we have

$$\mathbb{P}(|\hat{z}' - z| \geq \varepsilon z) \leq \delta/2.$$

---

[7] and of course the sampling complexity is $m'$ instead of $m$

Now let $p = (p_1, \ldots, p_n)$ be the probability distribution and let $\mathcal{C}$ denote the multi-set of indices sampled from $[n]$ when $m$ samples are taken from $[n]$ with respect to distribution $p$. For each index $j \in [n]$ let $U_j(m, p) = \mathbb{1}\left[j \in \mathcal{C}\right]$ be the indicator random variable of the event that index $j$ is sampled at least once and let $U(m, p) = \sum_{i=j}^n U_j(m, p)$. Note that $U$ is a random variable that denotes the number of unique samples that result from the sampling process described above, and its expectation is given by

$$
\begin{aligned}
\mathbb{E}\left[U(m, p)\right] &= \sum_{j=1}^n \mathbb{E}\left[U_j(m, p)\right] = \sum_{j=1}^n \mathbb{P}(i \in \mathcal{C}) \\
&= \sum_{j=1}^n \mathbb{P}(j \text{ is sampled at least once}) \\
&= \sum_{j=1}^n \left(1 - \mathbb{P}(j \text{ is not sampled})\right) \\
&= n - \sum_{j=1}^n (1 - p_j)^m.
\end{aligned}
$$

Now we want to establish the condition for which $U(m', q) < U(m, p)$, which, if it holds, would imply that the number of distinct weights that we retain with the deterministic + sampling approach is lower and still achieves the same error and failure probability guarantees, making it the overall better approach. To apply a strong concentration inequality, let $\mathcal{C}' = \mathcal{C}_{\text{det}} \cup \mathcal{C}_{\text{rand}} = \{c_1', \ldots, c_k', c_{k+1}', \ldots, c_{m'}'\}$ denote the set of indices sampled from the deterministic + sampling (with distribution $q$) approach, and let $\mathcal{C} = \{c_1, \ldots, c_m\}$ be the indices of the random samples obtained by sampling from distribution $p$. Let $f(c_1', \ldots, c_{m'}', c_1, \ldots, c_m)$ denote the difference $U(m', q) - U(m, p)$ in the number of unique samples in $\mathcal{C}'$ and $\mathcal{C}$. Note that $f$ satisfies the bounded difference inequality with Lipschitz constant 1 since changing the index of any single sample in $\mathcal{C} \cup \mathcal{C}'$ can change $f$ by at most 1. Moreover, there are $m' + m$ random variables, thus, applying McDiarmid's inequality (van Handel, 2014), we obtain

$$
\mathbb{P}(\mathbb{E}\left[U(m, p) - U(m', q)\right] - (U(m, p) - U(m', q)) \geq t) \leq \exp\left(\frac{-2t^2}{(m + m')}\right),
$$

this implies that for $t = \sqrt{\frac{\log(2/\delta)(m+m')}{2}}$,

$$
\mathbb{E}\left[U(m, p) - U(m', q)\right] \leq U(m, p) - U(m', q) + t
$$

with probability at least $1 - \delta/2$. Thus, this means that if $E[U(m, p)] - \mathbb{E}\left[U(m', q)\right] > t$, then $U(m, p) > U(m', q)$.

More specifically, recall that

$$
\mathbb{E}\left[U(m, p)\right] = n - \sum_{j=1}^n (1 - p_j)^m = n - \sum_{j=1}^n \left(1 - \frac{s_j}{S}\right)^m
$$

and

$$
\begin{aligned}
\mathbb{E}\left[U(m', q)\right] &= k + \sum_{j: q_j > 0} \left(1 - (1 - q_j)^{m'}\right) \\
&= k + (n - k) - \sum_{j: q_j > 0} (1 - q_j)^{m'} \\
&= n - \sum_{j: q_j > 0} (1 - q_j)^{m'} \\
&= n - \sum_{j \notin \mathcal{C}_{\text{det}}} \left(1 - \frac{s_j}{S - S_k}\right)^{m'}
\end{aligned}
$$

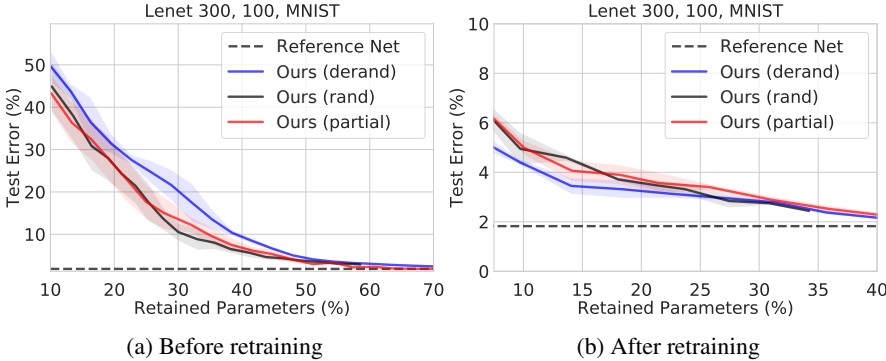

(a) Before retraining

(b) After retraining

Figure 5: The performance of our approach on a LeNet300-100 architecture trained on MNIST with no derandomization (denoted by "rand"), with partial derandomization (denoted by "partial"), and with complete derandomization (denoted by "derand"). The plot in (a) and (b) show the resulting test accuracy for various percentage of retained parameters $1 - (\text{pruneratio})$ before and after retraining, respectively. The additional error of the derandomized algorithm can be neglected in practical settings, especially after retraining.

Thus, rearranging terms, we conclude that it is better to conduct the deterministic + sampling scheme if

$$\sum_{j \notin \mathcal{C}_{\text{det}}} \left(1 - \frac{s_j}{S - S_k}\right)^{m'} > \sum_{j=1}^{n} \left(1 - \frac{s_j}{S}\right)^{m} + \sqrt{\frac{\log(2/\delta)(m + m')}{2}}.$$

Putting it all together, and conditioning on the above inequality holding, we have by the union bound

$$\mathbb{P}\left(|\hat{z}' - z| \geq \varepsilon z \cup U(m', q) > U(m, p)\right) \leq \delta,$$

this implies that with probability at least $1 - \delta$: (i) $\hat{z}' \in (1 \pm \varepsilon)z$ and (ii) $U(m', q) < U(m, p)$, implying that the deterministic + sampling approach ensures the error guarantee holds with a smaller number of unique samples, leading to better compression. □

### B.1.1 EXPERIMENTAL EVALUATION OF DERANDOMIZATION

To evaluate our theoretical results of derandomization, we tested the performance of our algorithm with respect to three different variations of sampling:

1. No derandomization ("rand"): We apply Alg. 2 and sample channels with probability proportional to their sensitivity.

2. Partial derandomization ("partial"): We apply Theorem 8 as a preprocessing step to keep the top $k$ channels and then sample from the rest according to Alg. 2.

3. Complete derandomization ("derand"): We simply keep the top channels until our sampling budget is exhausted.

The results of our evaluations on a LeNet300-100 architecture trained on MNIST can be seen in Fig. 5. As visible from Fig. 5(a), the process of partial derandomization does not impact the performance of our algorithm, while the complete derandomization of our algorithm has a slightly detrimental effect on the performance. This is in accordance to Theorem 8, which predicts that that it is best to only partially derandomize the sampling procedure. However, after we retrain the network, the additional error incurred by the complete derandomization is negligible as shown in Fig. 5(b). Moreover, it appears that – especially for extremely low sampling regime – the completely derandomized approach seems to incur a slight performance boost relative to the other approaches. We suspect that simply keeping the top channels may have a positive side effect on the optimization landscape during retraining, which we would like to further investigate in future research.

## C  MAIN COMPRESSION THEOREM

Having established layer-wise approximation guarantees as in Sec. B, all that remains to establish guarantees on the output of the entire network is to carefully propagate the error through the layers as was done in Baykal et al. (2019a). For each $i \in [\eta^{\ell+1}]$ and $\ell \in [L]$, define

$$\tilde{\Delta}_i^\ell(x) = {\left(z_i^+(x) + z_i^-(x)\right)}/{|z_i(x)|},$$

where $z_i^+(x) = \sum_{k \in I^+} w_{ik}^{\ell+1} a_k^\ell(\mathbf{x})$ and $z_i^-(x) = \sum_{k \in I^-} w_{ik}^{\ell+1} a_k^\ell(\mathbf{x})$ are positive and negative components of $z_i^{\ell+1}(x)$, respectively, with $I^+$ and $I^-$ as in Alg. 2. For each $\ell \in [L]$, let $\Delta^\ell$ be a constant defined as a function of the input distribution $\mathcal{D}$ [8], such that with high probability over $x \sim \mathcal{D}$, $\Delta^\ell \geq \max_{i \in [\eta^{\ell+1}]} \Delta_i^\ell$. Finally, let $\Delta^{\ell \rightarrow} = \prod_{k=\ell}^L \Delta^k$.

Generalizing Theorem 2 to obtain a layer-wise bound and applying error propagation bounds of Baykal et al. (2019a), we establish our main compression theorem below.

**Theorem 3.** *Let $\varepsilon, \delta \in (0,1)$ be arbitrary, let $\mathcal{S} \subset \mathcal{X}$ denote the set of $\lceil K' \log(4\eta/\delta) \rceil$ i.i.d. points drawn from $\mathcal{D}$, and suppose we are given a network with parameters $\theta = (W^1, \ldots, W^L)$. Consider the set of parameters $\hat{\theta} = (\hat{W}^1, \ldots, \hat{W}^L)$ generated by pruning channels of $\theta$ according to Alg. 2 for each $\ell \in [L]$. Then, $\hat{\theta}$ satisfies $\mathbb{P}_{\hat{\theta}, x \sim \mathcal{D}} \left( f_{\hat{\theta}}(x) \in (1 \pm \varepsilon) f_\theta(x) \right) \geq 1 - \delta$, and the number of filters in $\hat{\theta}$ is bounded by $\mathcal{O}\left( \sum_{\ell=1}^L \frac{L^2 (\Delta^{\ell \rightarrow})^2 S^\ell \log(\eta/\delta)}{\varepsilon^2} \right)$.*

## D  EXTENSION TO CNNS

To extend our algorithm to CNNs, we need to consider the fact that there is implicit weight sharing involved by definition of the CNN filters. Intuitively speaking, to measure the importance of a feature map (i.e. neuron) in the case of FNNs we consider the maximum impact it has on the preactivation $z^{\ell+1}(x)$. In the case of CNNs the same intuition holds, that is we want to capture the maximum contribution of a feature map $a_j^\ell(x)$, which is now a two-dimensional image instead of a scalar neuron, to the pre-activation $z^{\ell+1}(x)$ in layer $\ell + 1$. Thus, to adapt our algorithm to prune channels in CNNs, we modify the definition of sensitivity slightly, by also taking the maximum over the patches $p \in \mathcal{P}$ (i.e., sliding windows created by convolutions). In this context, each activation $a_j^\ell(x)$ is also associated with a patch $p \in \mathcal{P}$, which we denote by $a_{jp}^\ell$. In particular, the slight change is the following:

$$s_j^\ell = \max_{x \in \mathcal{S}} \max_{i \in [\eta^\ell]} \max_{p \in \mathcal{P}} \frac{w_{ij}^{\ell+1} a_{jp}^\ell(x)}{\sum_{k \in [\eta^\ell]} w_{ik}^{\ell+1} a_{kp}^\ell(x)},$$

where $a_{\cdot p}$ corresponds to the activation window associated with patch $p \in \mathcal{P}$. Everything else remains the same and the proofs are analogous.

## E  EXPERIMENTAL DETAILS AND ADDITIONAL EVALUATIONS

For our experimental evaluations, we considered a variety of data sets (MNIST, CIFAR-10, ImageNet) and neural network architectures (LeNet, VGG, ResNet, WideResNet, DenseNet) and compared against several state-of-the-art filter pruning methods. We conducted all experiments on either a single NVIDIA RTX 2080Ti with 11GB RAM or a NVIDIA Tesla V100 with 16GB RAM and implemented them in PyTorch (Paszke et al., 2017). Retraining with ImageNet was conducted on a cluster of 8 NVIDIA Tesla V100 GPUs.

In the following, we summarize our hyperparameters for training and give an overview of the comparison methods. All reported experimental quantities are averaged over three separately trained and pruned networks.

---

[8]If $\Delta_i(x)$ is a sub-Exponential random variable (Vershynin, 2016) with parameter $\lambda = O(1)$, then for $\delta$ failure probability: $\Delta^\ell \mathcal{O}(\mathbb{E}_{x \sim \mathcal{D}}[\max_i \Delta_i(x)] + \log(1/\delta))$ (Baykal et al., 2019a; Vershynin, 2016)

## E.1 COMPARISON METHODS

We further evaluated the performance of our algorithm against a variety of state-of-the-art methods in filter pruning as listed below. These methods were re-implemented for our own experiments to ensure an objective comparison method between the methods and we deployed the same iterative pruning and fine-tune strategy as is used in our method. Moreover, we considered a fixed pruning ratio of filters in each layers as none of the competing methods provide an automatic procedure to detect relative layer importance and allocate samples accordingly. Thus, the differentiating factor between the competing methods is their respective pruning step that we elaborate upon below.

**Filter Thresholding (FT, Li et al. (2016)** Consider the set of filters $W^\ell = [W_1^\ell, \ldots, W_{\eta^\ell}^\ell]$ in layer $\ell$ and let $\left\|W_j^\ell\right\|_{2,2}$ denote the entry-wise $\ell_2$-norm of $W_j^\ell$ (or Frobenius norm). Consider a desired sparsity level of $t\%$, i.e., we want to keep only $t\%$ of the filters. We then simply keep the filters with the largest norm until we satisfy our desired level of sparsity.

**SoftNet (He et al., 2018)** The pruning procedure of He et al. (2018) is similar in nature to the work of Li et al. (2016) except the saliency score used is the entrywise $\ell_1$ - norm $\left\|W_j^\ell\right\|_{1,1}$ of a filter map $W_j^\ell$. During their fine-tuning scheme they allow pruned filters to become non-zero again and then repeat the pruning procedure. As for the other comparisons, however, we only employ one-shot prune and fine-tune scheme.

**ThiNet (Luo et al., 2017)** Unlike the previous two approaches, which compute the saliency score of the filter $W_j^\ell$ by looking at its entry-wise norm, the method of Luo et al. (2017) iteratively and greedily chooses the feature map (and thus corresponding filter) that incurs the least error in an absolute sense in the pre-activation of the next layer. That is, initially, the method picks filter $j^*$ such that $j^* = \mathrm{argmin}_{j \in [\eta^\ell]} \max_{x \in \mathcal{S}} \left|z^{\ell+1}(x) - z_{[j]}^{\ell+1}(x)\right|$, where $z^{\ell+1}(x)$ denotes the pre-activation of layer $\ell + 1$ for some input data point $x$, $z_{[j]}^{\ell+1}(x)$ the pre-activation when only considering feature map $j$ in layer $\ell$, and $\mathcal{S}$ a set of input data points. We note that this greedy approach is quadratic in both the size $\eta^\ell$ of layer $\ell$ and the size $|\mathcal{S}|$ of the set of data points $\mathcal{S}$, thus rendering it very slow in practice. In particular, we only use a set $\mathcal{S}$ of cardinality comparable to our own method, i.e., around 100 data points in total. On the other hand, Luo et al. report to use 100 data points per output class resulting in 1000 data points for CIFAR10.

## E.2 LENET ARCHITECTURES ON MNIST

We evaluated the performance of our pruning algorithm and the comparison methods on LeNet300-100 (LeCun et al., 1998), a fully-connected network with two hidden layers of size 300 and 100 hidden units, respectively, and its convolutional counterpart, LeNet-5 (LeCun et al., 1998), which consists of two convolutional layers and two fully-connected layers. Both networks were trained on MNIST using the hyper-parameters specified in Table 3. We trained on a single GPU and during retraining (fine-tunine) we maintained the same hyperparameters and only adapted the ones specifically mentioned in Table 3.

| | | LeNet-300-100 | LeNet-5 |
|---|---|---|---|
| | test error | 1.59 | 0.72 |
| | loss | cross-entropy | cross-entropy |
| | optimizer | SGD | SGD |
| | epochs | 40 | 40 |
| Train | batch size | 64 | 64 |
| | LR | 0.01 | 0.01 |
| | LR decay | 0.1@{30} | 0.1@{25, 35} |
| | momentum | 0.9 | 0.9 |
| | weight decay | 1.0e-4 | 1.0e-4 |
| Prune | $\delta$ | 1.0e-12 | 1.0e-12 |
| | $\alpha$ | not iterative | 1.18 |
| Fine-tune | epochs | 30 | 40 |
| | LR decay | 0.1@{20, 28} | 0.1@{25, 35} |

Table 3: We report the hyperparameters used during MNIST training, pruning, and fine-tuning for the LeNet architectures. LR hereby denotes the learning rate and LR decay denotes the learning rate decay that we deploy after a certain number of epochs. During fine-tuning we used the same hyperparameters except for the ones indicated in the lower part of the table.

We further evaluated the performance of our algorithm on a variety of convolutional neural network architectures trained on CIFAR-10. Specifically, we tested it on VGG16 with batch norm (Simonyan & Zisserman, 2014), ResNet20 (He et al., 2016), DenseNet22 (Huang et al., 2017), and Wide ResNet-16-8 (Zagoruyko & Komodakis, 2016). For residual networks with skip connections, we model the interdependencies between the feature maps and only prune a feature map if it does not get used as an input in the subsequent layers. We performed the training on a single GPU using the same hyperparameters specified in the respective papers for CIFAR training. During fine-tuning we kept the number of epochs and also did not adjusted the learning rate schedule. We summarize the set of hyperparameters for the various networks in Table 4.

|  |  | VGG16 | ResNet20/56/110 | DenseNet22 | WRN-16-8 |
|---|---|---|---|---|---|
| Train | test error | 7.11 | 8.59/7.05/6.43 | 10.07 | 4.81 |
|  | loss | cross-entropy | cross-entropy | cross-entropy | cross-entropy |
|  | optimizer | SGD | SGD | SGD | SGD |
|  | epochs | 300 | 182 | 300 | 200 |
|  | batch size | 256 | 128 | 64 | 128 |
|  | LR | 0.05 | 0.1 | 0.1 | 0.1 |
|  | LR decay | $0.5@\{30,\dots\}$ | $0.1@\{91, 136\}$ | $0.1@\{150, 225\}$ | $0.2@\{60,\dots\}$ |
|  | momentum | 0.9 | 0.9 | 0.9 | 0.9 |
|  | Nesterov | ✗ | ✗ | ✓ | ✓ |
|  | weight decay | 5.0e-4 | 1.0e-4 | 1.0e-4 | 5.0e-4 |
| Prune | $\delta$ | 1.0e-16 | 1.0e-16 | 1.0e-16 | 1.0e-16 |
|  | $\alpha$ | 1.50 | 0.50/0.79/0.79 | 0.40 | 0.36 |
| Fine-tune | epochs | 150 | 182 | 300 | 200 |

Table 4: We report the hyperparameters used during training, pruning, and fine-tuning for various convolutional architectures on CIFAR-10. LR hereby denotes the learning rate and LR decay denotes the learning rate decay that we deploy after a certain number of epochs. During fine-tuning we used the same hyperparameters except for the ones indicated in the lower part of the table. $\{30,\dots\}$ denotes that the learning rate is decayed every 30 epochs.

### E.4 Convolutional Neural Networks on ImageNet

We consider pruning convolutional networks of varying size – ResNet18, ResNet50, and ResNet101 – trained on the ImageNet (Russakovsky et al., 2015) data set. The hyperparameters used for training and for our pruning algorithm are shown in Table 5. For this dataset, we considered two scenarios: (i) iterative pruning without retraining and (ii) iterative prune-retrain with a limited amount of iterations given the resource-intensive nature of the experiments.

In the first scenario, we evaluate the baseline effectiveness of each pruning algorithm *without fine-tuning* by applying the same iterative prune-scheme, but without the retraining step. The results of these evaluations can be seen in Fig. 6. Fig. 6 shows that our algorithm outperforms the competing approaches in generating compact, more accurate networks. We suspect that by reevaluating the data-informed filter importance (empirical sensitivity) after each iteration our approach is capable of more precisely capturing the inter-dependency between layers that alter the relative importance of filters and layers with each pruning step. This is in contrast to competing approaches, which predominantly rely on weight-based criteria of filter importance, and thus can only capture this inter-dependency after retraining (which subsequently alters the magnitude of the weights).

|  |  | ResNet18/50/101 |
|---|---|---|
| Train | top-1 test error | 30.26/23.87/22.63 |
|  | top-5 test error | 10.93/7.13/6.45 |
|  | loss | cross-entropy |
|  | optimizer | SGD |
|  | epochs | 90 |
|  | batch size | 256 |
|  | LR | 0.1 |
|  | LR decay | $0.1@\{30, 60\}$ |
|  | momentum | 0.9 |
|  | Nesterov | ✗ |
|  | weight decay | 1.0e-4 |
| Prune | $\delta$ | 1.0e-16 |
|  | $\alpha$ | 0.43/0.50/0.50 |
| Fine-tune | epochs | 90 |

Table 5: The hyper-parameters used for training and pruning residual networks trained on the ImageNet data set.

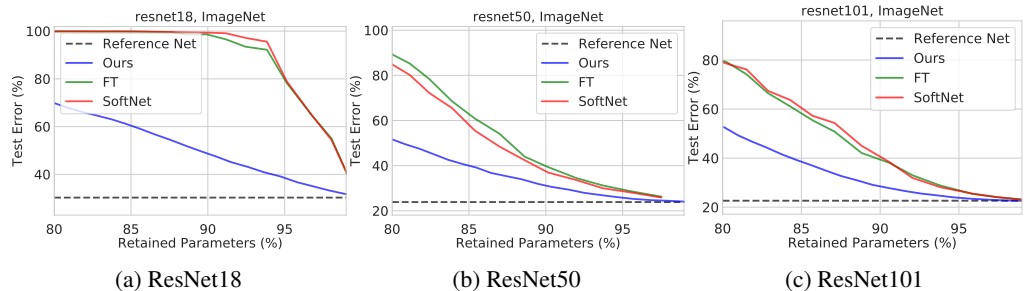

| | (a) ResNet18 | (b) ResNet50 | (c) ResNet101 |

Figure 6: The results of our evaluations of the algorithms in the *prune-only* scenario, where the network is iteratively pruned down to a specified target prune ratio and the fine-tuning step is omitted. Note that the $x$ axis is the percentage of parameters retained, i.e., $(1 - \text{pruneratio})$.

| Model | Method | Top-1 Err. (%) | | | Top-5 Err. (%) | | | PR (%) | FR (%) |
|---|---|---|---|---|---|---|---|---|---|
| | | Orig. | Pruned | Diff. | Orig. | Pruned | Diff. | | |
| | Ours (within 4.0% top-1) | 30.26 | 34.35 | +4.09 | 10.93 | 13.25 | +2.32 | **60.48** | **43.12** |
| | Ours (within 2.0% top-1) | 30.26 | 32.62 | +2.36 | 10.93 | 12.09 | +1.16 | 43.80 | 29.30 |
| | Ours (lowest top-1 err.) | 30.26 | **31.34** | **+1.08** | 10.93 | **11.43** | **+0.50** | 31.03 | 19.99 |
| Resnet18 | He et al. (2018) (SoftNet) | 29.72 | 32.90 | +3.18 | 10.37 | 12.22 | +1.85 | N/A | 41.80 |
| | He et al. (2019) | 29.72 | 31.59 | +1.87 | 10.37 | 11.52 | +1.15 | N/A | 41.80 |
| | Dong et al. (2017) | 30.02 | 33.67 | +3.65 | 10.76 | 13.06 | +2.30 | N/A | 33.30 |
| | Ours (within 1.0% top-1) | 23.87 | 24.79 | +0.92 | 7.13 | 7.57 | +0.45 | **44.04** | 30.05 |
| | Ours (lowest top-1 err.) | 23.87 | **24.09** | **+0.22** | 7.13 | **7.19** | **+0.06** | 18.01 | 10.82 |
| | He et al. (2018) (SoftNet) | 23.85 | 25.39 | +1.54 | 7.13 | 7.94 | +0.81 | N/A | 41.80 |
| Resnet50 | Luo et al. (2017) (ThiNet) | 27.12 | 27.96 | +0.84 | 8.86 | 9.33 | +0.47 | 33.72 | 36.39 |
| | He et al. (2019) | 23.85 | 25.17 | +1.32 | 7.13 | 7.68 | +0.55 | N/A | 53.50 |
| | He et al. (2017) | N/A | N/A | N/A | 7.80 | 9.20 | +1.40 | N/A | 50.00 |
| | Luo & Wu (2018) | 23.85 | 25.24 | +1.39 | 7.13 | 7.85 | +0.72 | N/A | 48.70 |
| | Liu et al. (2019a) | 23.40 | 24.60 | +1.20 | N/A | N/A | N/A | N/A | **51.22** |
| | Ours (within 1.0% top-1) | 22.63 | 23.57 | +0.94 | 6.45 | 6.89 | +0.44 | **50.45** | **45.08** |
| | Ours (lowest top-1 err.) | 22.63 | 23.22 | +0.59 | 6.45 | 6.74 | +0.29 | 33.04 | 29.38 |
| Resnet101 | He et al. (2018) (SoftNet) | 22.63 | **22.49** | **-0.14** | 6.44 | **6.29** | **-0.15** | N/A | 42.20 |
| | He et al. (2019) | 22.63 | 22.68 | +0.05 | 6.44 | 6.44 | +0.00 | N/A | 42.20 |
| | Ye et al. (2018) | 23.60 | 24.73 | +1.13 | N/A | N/A | N/A | 47.20 | 42.69 |

Table 6: Comparisons of the performance of various pruning algorithms on ResNets trained on ImageNet (Russakovsky et al., 2015). The reported results for the competing algorithms were taken directly from the corresponding papers. For each network architecture, the best performing algorithm for each evaluation metric, i.e., Pruned Err., Err. Diff, PR, and FR, is shown in bold.

Next, we consider pruning the networks using the standard iterative prune-retrain procedure as before (see Sec. E) with only a limited number of iterations (2-3 iterations per reported experiment). The results of our evaluations are reported in Table 6 with respect to the following metrics: the resulting error of the pruned network (Pruned Err.), the difference in model classification error (Err. Diff), the percentage of parameters pruned (PR), and the FLOP Reduction (FR). We would like to highlight that – despite the limited resources used during the experiments – our method is able to produce compressed networks that are as accurate and compact as the models generated by competing approaches (obtained by significantly more prune-retrain iterations than allotted to our algorithm).

## E.5 APPLICATION TO REAL-TIME REGRESSION TASKS

In the context of autonomous driving and other real-time applications of neural network inference, fast inference times while maintaining high levels of accuracy are paramount to the successful deployment of such systems (Amini et al., 2018). The particular challenge of real-time applications stems from

the fact that – in addition to the conventional trade-off between accuracy and model efficiency – inference has to be conducted in real-time. In other words, there is a hard upper bound on the allotted computation time before an answer needs to be generated by the model. Model compression, and in particular, filter compression can provide a principled approach to generating high accuracy outputs without incurring high computational cost. Moreover, the provable nature of our approach is particularly favorable for real-time applications, as they usually require extensive performance guarantees before being deployed, e.g., autonomous driving tasks.

To evaluate the empirical performance of our filter pruning method on real-time systems, we implemented and tested the neural network of Amini et al. (2018), which is a regression neural network deployed on an autonomous vehicle in real time to predict the steering angle of the human driver. We trained the network of Amini et al. (2018), denoted by *Deepknight*, with the driving data set provided alongside, using the hyperparameters summarized in Table 7.

The results of our compression can be found in Fig. 7, where we evaluated and compared the performance of our algorithm to those of other SOTA methods (see Sec. E.6). We note that these results were achieved *without* retraining as our experimental evaluations have shown that even without retraining we can achieve significant pruning ratios that lead to computational speed-ups in practice. As apparent from Fig. 7, we can again outperform other SOTA methods in terms of performance vs. prune ratio. Note that since this is a regression task, we used test loss (mean-squared error on the test data set) as performance criterion.

|  |  | Deepknight |
|---|---|---|
| | test loss | 4.9e-5 |
| | loss | MSE |
| | optimizer | Adam |
| | epochs | 100 |
| Train | batch size | 32 |
| | LR | 1e-4 |
| | LR decay | 0.1@{50, 90} |
| | momentum | 0.9 |
| | weight decay | 1.0e-4 |
| Prune | $\delta$ | 1.0e-32 |
| | $\alpha$ | not iterative |
| Fine-tune | epochs | 0 |

Table 7: We report the hyperparameters used for training and pruning the driving network of Amini et al. (2018) together with the provided data set. No fine-tuning was conducted for this architecture. LR hereby denotes the learning rate, LR decay denotes the learning rate decay that we deploy after a certain number of epochs, and MSE denotes the mean-squared error.

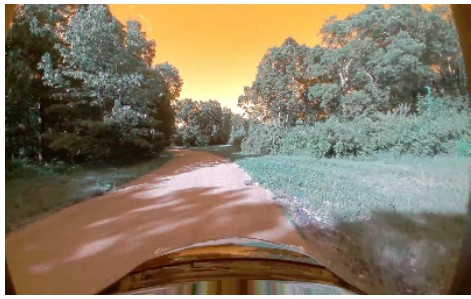

(a) Example driving image from Amini et al. (2018)

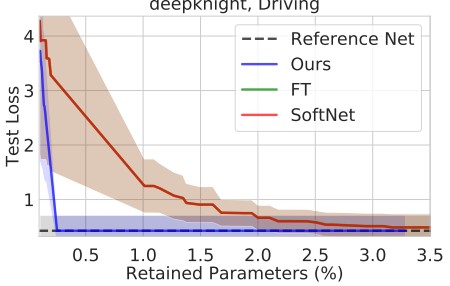

(b) Pruning performance before retraining

Figure 7: The performance of our approach on a regression task used to infer the steering angle for an autonomous driving task (Amini et al., 2018). (a) An exemplary image taken from the data set. (b) The performance of our pruning procedure before retraining evaluated on the test loss and compared to competing filter pruning methods. Note that the $x$ axis is percentage of parameters retained, i.e., $1 - (\text{pruneratio})$.

Finally, we would like to highlight that our filter pruning method may serve as a principled subprocedure during the design of neural network architectures for real-time applications. In particular, given an inference time budget $\mathcal{T}$, one can design and train much larger architectures with favorable performance which, however, violate the given budget $\mathcal{T}$. Our filter pruning method can then be leveraged to compress the network until the given budget $\mathcal{T}$ is satisfied, thus reducing the burden on the practitioner to design a simultaneously accurate and computationally-efficient neural network architecture.

### E.6 COMPARISONS TO ADDITIONAL METHODS ON CIFAR-10

We evaluate the performance of our algorithm on pruning modern convolutional and residual bench-mark network architectures – ResNet20, ResNet56, ResNet110, and VGG16 – trained on CIFAR-10, and compare it to the results reported by state-of-the-art filter pruning approaches. The results are obtained by iteratively pruning and fine-tuning the resulting model (starting from the original pre-trained network) in a hyperharmonic sequence of prune ratios as described in Sec. 4.1. For our algorithm, in addition to reporting the pruned model that achieves commensurate accuracy ("within $0.5\%$ err.") as in Sec. 4, we report the model that (i) closest matches the accuracy of the original network ("orig. err.") and (ii) achieves the lowest classification error possible ("lowest err.") – which for nearly all of the models considered, is lower than that of the original model.

Table 8 summarizes our results and depicts the performance of each pruning algorithm with respect to various metrics: the resulting error of the pruned network (Pruned Err.), the difference in model classification error (Err. Diff), the percentage of parameters pruned (PR), and the FLOP Reduction (FR). Our results show that our algorithm outperforms competing approaches in virtually all of the considered models and pertinent metrics, especially when the overall quality of the pruned model is taken into account.

For ResNet20 for instance, our algorithm generates a model that is simultaneously the sparsest ($>43\%$ PR) and the most accurate ($8.64\%$ Err., $0.04\%$ Err. Diff) *despite* starting from a pre-trained model with the highest error of $8.60\%$ (Orig. Err.) among the reported results. The method of He et al. (2019) does achieve a higher FR than our method, however, this is achieved at the cost of nearly $2\%$ degradation in classification accuracy (compared to $0.04\%$ for ours).

For larger networks such as ResNet110, our algorithm's favorable performance is even more pro-nounced: the models generated by our algorithm are not only the sparsest (PR) and most efficient (PR), but they are also the most accurate. A nearly identical trend holds for the results pertaining to VGG16 and ResNet56: the models generated by our method tend to be the overall sparsest and most accurate, even when starting from pre-trained models with higher classification error.

| Model | Method | Orig. Err. (%) | Pruned Err. (%) | Err. Diff. (%) | PR (%) | FR (%) |
|---|---|---|---|---|---|---|
| | Ours (within 0.5% err.) | 8.60 | 9.09 | +0.49 | **62.67** | 45.46 |
| | Ours (orig. err.) | 8.60 | **8.64** | **+0.04** | 43.16 | 32.10 |
| | Ours (lowest err.) | 8.60 | **8.64** | **+0.04** | 43.16 | 32.10 |
| ResNet20 | He et al. (2018) (SoftNet) | 7.80 | 8.80 | +1.00 | N/A | 29.30 |
| | He et al. (2019) | 7.80 | 9.56 | +1.76 | N/A | **54.00** |
| | Ye et al. (2018) | 8.00 | 9.10 | +1.10 | 37.22 | N/A |
| | Lin et al. (2020) | 7.52 | 9.72 | +2.20 | 40.00 | N/A |
| | Ours (within 0.5% err.) | 7.05 | 7.33 | +0.28 | **88.98** | **84.42** |
| | Ours (orig. err.) | 7.05 | 7.02 | -0.03 | 86.00 | 80.76 |
| | Ours (lowest err.) | 7.05 | 6.36 | **-0.69** | 72.10 | 67.41 |
| | Li et al. (2016) (FT) | 6.96 | 6.94 | -0.02 | 13.70 | 27.60 |
| ResNet56 | He et al. (2018) (SoftNet) | 6.41 | 6.65 | +0.24 | N/A | 52.60 |
| | He et al. (2019) | 6.41 | 6.51 | +0.10 | N/A | 52.60 |
| | He et al. (2017) | 7.20 | 8.20 | +1.00 | N/A | 50.00 |
| | Li et al. (2019) | 6.28 | 6.60 | +0.32 | 78.10 | 50.00 |
| | Lin et al. (2020) | 5.49 | **5.97** | +0.48 | 40.00 | N/A |
| | Ours (within 0.5% err.) | 6.43 | 6.79 | +0.36 | **92.07** | **89.76** |
| | Ours (orig. err.) | 6.43 | 6.35 | -0.08 | 89.15 | 86.97 |
| | Ours (lowest err.) | 6.43 | **5.42** | **-1.01** | 71.98 | 68.94 |
| ResNet110 | Li et al. (2016) (FT) | 6.47 | 6.70 | +0.23 | 32.40 | 38.60 |
| | He et al. (2018) (SoftNet) | 6.32 | 6.14 | -0.18 | N/A | 40.80 |
| | He et al. (2019) | 6.32 | 6.16 | -0.16 | N/A | 52.30 |
| | Dong et al. (2017) | 6.37 | 6.56 | +0.19 | N/A | 34.21 |
| | Ours (within 0.5% err.) | 7.28 | 7.78 | +0.50 | **94.32** | **85.03** |
| | Ours (orig. err.) | 7.28 | 7.17 | -0.11 | 87.06 | 70.32 |
| | Ours (lowest err.) | 7.28 | 7.06 | **-0.22** | 80.02 | 59.21 |
| VGG16 | Li et al. (2016) (FT) | 6.75 | 6.60 | -0.15 | 64.00 | 34.20 |
| | Huang et al. (2018) | 7.23 | 7.83 | +0.60 | 83.30 | 45.00 |
| | He et al. (2019) | 6.42 | 6.77 | +0.35 | N/A | 35.90 |
| | Li et al. (2019) | 5.98 | **6.18** | +0.20 | 78.20 | 76.50 |

Table 8: The performance of our algorithm and that of state-of-the-art filter pruning algorithms on modern CNN architectures trained on CIFAR-10. The reported results for the competing algorithms were taken directly from the corresponding papers. For each network architecture, the best performing algorithm for each evaluation metric, i.e., Pruned Err., Err. Diff, PR, and FR, is shown in **bold**. The results show that our algorithm consistently outperforms state-of-the-art pruning approaches in nearly all of the relevant pruning metrics.

