# OpenReview forum: "Provable Filter Pruning for Efficient Neural Networks"
_ICLR.cc/2020/Conference — Accept (Poster)_

### Official Review · AnonReviewer2 · 2019-10-25
**Official Blind Review #2**

**Rating:** 3

**Review:**

This paper studies the tasks of pruning filters, a provable, sampling-based approach for generating compact Convolutional Neural Networks (CNNs). This paper gives rise to a fully-automated procedure for identifying and preserving the filters in layers that are essential to the network’s performance.  In general, this paper is very well written and organized.

1, The key concerns come from the Lottery papers [1,2]. One can find sparse structure from an overparameterized model.  The results of pruned network should be improved, rather than getting worse, since some redundant filters/params are removed from original network. In contrast, all the results of this method gets worse results; this is less desirable.

2, the theoretical analysis is very good;  it is worthy publishing themselves. But ever since the "lottery" papers, I think it makes sense in locating the sparse and representative pruned structure, which can achieve better performance than full overparameterized model.
so it’s quite a borderline paper.


[1] THE LOTTERY TICKET HYPOTHESIS: FINDING SPARSE, TRAINABLE NEURAL NETWORKS. ICLR 2019.
[2] RETHINKING THE VALUE OF NETWORK PRUNING. ICLR 2019.



**Experience Assessment:**

I have published one or two papers in this area.

**Review Assessment: Checking Correctness Of Derivations And Theory:**

I assessed the sensibility of the derivations and theory.

**Review Assessment: Checking Correctness Of Experiments:**

I assessed the sensibility of the experiments.

**Review Assessment: Thoroughness In Paper Reading:**

I read the paper at least twice and used my best judgement in assessing the paper.

---

> ### Author Response · Authors · 2019-11-12
> **Response to AnonReviewer2**
>
> Thank you for your consideration of our paper and for your thoughtful comments. We would like to first clarify that the Lottery Ticket Hypothesis [1] does not claim that the predictive accuracy of the pruned network is necessarily better than the unpruned network (pg. 2, para. 2 of [1]). In fact, from the figures of [1] (e.g., Fig. 3), we can see that the performance of the pruned network may improve at moderate sparsities (i.e., pruning ratios). On the other hand, in the regime of extreme sparsities the significantly compactified network may not perform as well as the original network [1]. This makes intuitive sense, since if it were the case that the pruned network’s performance simply improves as we prune-and-retrain iteratively, we would end up with pruned networks with only a handful of parameters that outperform the original network.
>
> This phenomenon occurs for, e.g., ResNet56, ResNet110, VGG16, and DenseNet22 in Fig. 3 of our revision. The unifying trend in all of these plots is that the pruned model tends to outperform the original model in the regime of moderate sparsities, however, as we compress the model further down and consider higher sparsities, we see that the pruned model’s performance can no longer match (or outperform) the original model’s performance.
>
> The reported test error of the pruned model in Tables 1 and 2 of our submission are slightly above that of the original model because these tables focused on reporting the sparsest model possible that was within 0.5% of the original model’s accuracy (i.e., commensurate accuracy). To further contextualize the performance of our method, we have added an additional table (Table 8 in the appendix) that reports statistics pertaining to the pruned model that (i) is within 0.5% of the original model’s accuracy (akin to Tables 1 and 2), (ii) matches the original accuracy, and (iii) achieves the highest accuracy possible -- which in some cases, is higher than that of the original model. As our results show, the trade-off between predictive accuracy of the model and its size is an application-dependent choice that is up to the practitioner to decide.
>
> [1] The Lottery Ticket Hypothesis: Finding Sparse, Trainable Neural Networks (https://arxiv.org/pdf/1803.03635.pdf)

---

### Official Review · AnonReviewer1 · 2019-10-25
**Official Blind Review #1**

**Rating:** 6

**Review:**

This paper attacks the problem of pruning neural networks to obtain sparser models for deployment. In introduces a principled importance sampling approach for which independence of samples allows one to obtain bounds easily. These bounds can be used to control the accuracy of the method.

The proposal mechanism is very smart. The authors use a measure of the sensitivity of the network outputs to the channels in a particular layer (eqn 1).

The paper is very well written, but it would help to add a picture where all the symbols in section 2.1 appear. At times it is hard to keep track of the channels and features. It might alternatively be a good idea to specify the equation of a layer (eg what eventually ends up happening at the bottom of page 3) in section 2.1 and then explain the symbols in the equation. This will make life easier for anyone reading the paper for the first time.

The experiments are well execute and include reasonable baselines. I would in addition recommend this recent paper:
https://arxiv.org/abs/1902.09574

It would also be nice to relate the work to this best paper this year.
https://openreview.net/forum?id=rJl-b3RcF7

**Experience Assessment:**

I have published one or two papers in this area.

**Review Assessment: Checking Correctness Of Derivations And Theory:**

I assessed the sensibility of the derivations and theory.

**Review Assessment: Checking Correctness Of Experiments:**

I assessed the sensibility of the experiments.

**Review Assessment: Thoroughness In Paper Reading:**

I read the paper at least twice and used my best judgement in assessing the paper.

---

> ### Author Response · Authors · 2019-11-12
> **Response to AnonReviewer1**
>
> Thank you for your in-depth review of our paper and helpful suggestions in improving the clarity and quality of our work. Our revision contains a streamlined and improved Sec. 2 that includes further clarifying text — such as the equation of a layer — and additional explanation of the symbols in Sec. 2.1. We have also improved Fig. 1 and have made explicit references to it in Sec. 2 in order to better illustrate the denotations of the symbols used and the pruning pipeline.
>
> We also thank the reviewer for the references to related work. We have added additional text in the introduction and related works sections of our paper that further relate back to the work of [1] and [2].
>
> [1] The Lottery Ticket Hypothesis: Finding Sparse, Trainable Neural Networks (https://arxiv.org/pdf/1803.03635.pdf)
> [2] The State of Sparsity in Deep Neural Networks
> (https://arxiv.org/abs/1902.09574)

---

### Official Review · AnonReviewer5 · 2019-11-02
**Official Blind Review #1**

**Rating:** 3

**Review:**

Summary:

In this paper, the author propose a provable pruning method, and also provide a bound for the final pruning error.  Among most heuristics prune method,  pruning with mathematics guarantee is indeed more convincing. We expect this work can help people devoting some effort into more solid theoretical study in understanding the over-parameterized training.

Intuitively speaking, the sensitive neuron has greater contribution for the final output, reusing the corresponding filter and carefully rescale its value require many empirically attempts. To achieve a more reasonable algorithm, author prune the redundant channel by controlling the deviation of the summation statistically small,  and reusing the filter by important sampling the given channel. Experiment show that this method can reach a competitive prune radio against other pruning algorithm, and show robustly in retained parameters vs error experiment.


Weakness:

1. experiment is too weak

ImageNet model has great impact on most CV problem, and the current release models are flooding in the open source world. Author should at least provide a imagenet model and make this work more convincing. Besides, Author should also consider an experiment in modern lightweight network, vgg and resnet like model are out of fashion and so big that any one can make a sound result on it.

2. lack of a comparing experiment for random select the top-k norm.

Important sampling require an input of probability [p1, p2, p3, ... pn],  if those probabilities are nearly uniform, important sampling will behave like a random sampling method. In most case, if we want to prune the large channel network,  picking the top-1 significant filter or random sampling top-k filter will almost do the same thing.

3. lack of further theory consideration

author only consider the single layer reconstruction, without discussing the overall accumulative error. Unlike the other deterministic method, sampling skill suffer variance propagation problem, the pre-layer variance will affect the sampling probability of next layer, how this pruning work if we change status of the pre-layer,  I didn't find any theoretical guarantee and only find a proof of single layer reconstruction bound.


**Experience Assessment:**

I have read many papers in this area.

**Review Assessment: Checking Correctness Of Derivations And Theory:**

I assessed the sensibility of the derivations and theory.

**Review Assessment: Checking Correctness Of Experiments:**

I assessed the sensibility of the experiments.

**Review Assessment: Thoroughness In Paper Reading:**

I read the paper at least twice and used my best judgement in assessing the paper.

---

> ### Author Response · Authors · 2019-11-12
> **Response to AnonReviewer5**
>
> Thank you for your insightful and constructive feedback. Please find our specific comments below.
>
> 1)
> Our initial experimental results were computed on a desktop computer to highlight that the method we propose is practical and easy to use for a wide range of data sets and network models. Since the initial submission, we have conducted more extensive experiments on the Cloud with a refined iterative prune-retrain strategy including larger models for CIFAR10 (see Fig. 3 and Tables 2 and 8 of our revision). We are currently running experiments on the ImageNet data set evaluated on various ResNet architectures.
>
> Our preliminary results (single prune-retrain cycle) of the experiments conducted on ResNet101 show that we can prune 55.40% (PR) of the network and achieve a FLOP Reduction (FR) of 50.80% with only a 0.69% increase in the Top-5 classification error on ImageNet. We are currently running refined ImageNet experiments with iterative prune-retrain cycles (as outlined in Sec. 4.1) and we will update our submission within the next couple of days with our results. We would also like to note that VGG and ResNets are commonly used for baseline evaluations and comparisons in contemporary network pruning literature [1-3]. Please see our general response for a list of updates to the results section.
>
> The network used for the real-time regression task (see Sec. 4.5) of [4] is a lightweight model for use in autonomous driving scenarios. Our paper contains evaluations of our compression algorithm on this lightweight network (see Fig. 4). We would be happy to include results for other (lightweight) models as time permits. Please let us know if you have any particular models in mind for evaluation.
>
> 2)
> We have clarified the exposition to highlight that our original (and revised) submission already contains comparisons for selecting the top-k filters with the highest norms. The Filter Thresholding (FT) and SoftNet algorithms that are plotted in the figures and reported in the tables in Sec. 4 correspond precisely to this approach (also see Sec. E.1 - Comparison Methods of the appendix). In particular, the FT algorithm picks the k filters with the largest \ell_2-norm (i.e., top-k \ell_2 norm), whereas SoftNet picks the k filters with the largest \ell_1 norm.
>
> 3)
> Our main compression theorem (Theorem 8 in the appendix of the original submission) establishes bounds on the overall accumulative error, not just layer-wise error. The proof of the error propagation through all of the layers follows from the application of our layer-wise error bound (Theorem 2 of Sec. 2) and the error-propagation analysis of [4] (see Lemmas 2,3 and Theorem 4 in [4]). We agree with the reviewer that the reconstruction error of the previous layer has an affect on the variance of our estimator in the subsequent layers, however, this is taken care of in the error propagation analysis by iteratively conditioning on the fact that the previous layer is well-approximated, implying that our bounds are not affected by more than a constant.
>
> Nevertheless, we understand that the placement of Theorem 8 in the appendix may have caused some unintended confusion regarding the scope of our theoretical guarantees — which, to be clear, are not just layer-wise guarantees, and in fact do ensure that the output of the network is well-preserved. To minimize confusion, we have moved our main compression theorem (Theorem 8) to the main body of the paper (see Theorem 3 of our revision) and have made the application of the error propagation bounds explicit. Thank you again for pointing out this source of confusion.
>
>
> [1] The Lottery Ticket Hypothesis: Finding Sparse, Trainable Neural Networks (https://arxiv.org/pdf/1803.03635.pdf)
> [2] SNIP: Single-shot Network Pruning Based on Connection Sensitivity ( https://arxiv.org/pdf/1810.02340.pdf )
> [3] Importance Estimation for Neural Network Pruning (http://openaccess.thecvf.com/content_CVPR_2019/papers/Molchanov_Importance_Estimation_for_Neural_Network_Pruning_CVPR_2019_paper.pdf)
> [4] Data-Dependent Coresets for Compressing Neural Networks with Applications to Generalization Bounds (https://arxiv.org/abs/1804.05345)

---

### Author Response · Authors · 2019-11-12
**General Response**

We thank all the reviewers for their careful consideration of our paper and helpful suggestions. We have submitted a revised version that contains an improved exposition of our work thanks to the reviewers’ constructive feedback. Since our original submission, we have also conducted and included additional empirical evaluations in our revision. In particular, we:

1) re-ran our experiments with a refined, iterative prune-retrain scheme — as is standard in literature — with a larger number of retraining epochs,

2) included additional evaluations and figures with VGG16 with Batch Normalization, ResNet56, and ResNet110 (please see updated Fig. 3 and Table 2), and

3) added a new table to the appendix (Table 8, Sec. E.6) containing extensive evaluations and comparisons to state-of-the-art pruning results (as reported in the respective papers) published within the last couple of years. The latest results show that our approach outperforms competing filter pruning methods in virtually all of the considered network architectures and pertinent metrics, especially when the overall quality (i.e., sparsity, efficiency, and accuracy) of the pruned model is taken into account.

Moreover, we are currently running experiments on various ResNet architectures trained on ImageNet and will upload another revision within the next couple of days. In addition to conducting and including the results of additional empirical evaluations, we have also refined the presentation of our paper for ease of readability and understanding. Overall, we believe that the quality and exposition of our paper have significantly improved thanks to the reviewers’ suggestions and we welcome any additional feedback. We hope that the results in this paper, pruning with performance guarantees backed by extensive experiments, highlight that we can improve the efficiency and storage requirements of modern neural networks in a practical and theoretically-grounded way.

---

### Author Response · Authors · 2019-11-15
**General Response -- ImageNet Results**

We have updated our paper with a revised version containing the results of our ImageNet evaluations. We believe that the new results highlight the versatility of our approach in that it is readily applicable to large-scale problems out-of-the-box. In other words, our new results on ImageNet were obtained by running our algorithm without any tuning of the hyper-parameters; this stands in contrast to existing approaches that generally require tedious, task-specific intervention or manual parameter tuning [1-2].

More specifically, we evaluated and compared our algorithm in two scenarios on ResNet18/50/101 models trained on ImageNet: (i) prune-only experiments where the retraining step is omitted after pruning (see Fig. 6 in Sec. E.4) and (ii) iterative prune-retrain with a limited number (i.e., 2-3) of iterations (see Table 6 in Sec. E.4). For the former prune-only scenario, our algorithm significantly outperformed competing pruning approaches in all of the considered models. This suggests that our algorithm’s baseline pruning effectiveness is better than that of the competing methods.

For the latter prune-retrain scenario (with limited iterations), we observe from Table 6 that our approach is competitive with the results obtained by the various state-of-the-art pruning algorithms. We would like to highlight that our algorithm’s performance was competitive despite significant time (2-3 days) and resource constraints (8 NVIDIA Tesla V100). We conjecture that given more time and resources to conduct additional train-prune iterations for ImageNet, the relative performance of our algorithm would be even more favorable and would resemble the comparisons on CIFAR10 (see Table 8).


[1] Pruning Filters for Efficient Convnets
(https://arxiv.org/abs/1608.08710)
[2] Soft Filter Pruning for Accelerating Deep Convolutional Neural Networks
(https://arxiv.org/abs/1808.06866)

---

### Decision · Program_Chairs · 2019-12-19

**Decision:**

Accept (Poster)

**Comment:**

This paper presents a sampling-based approach for generating compact CNNs by pruning redundant filters. One advantage of the proposed method is a bound for the final pruning error.

One of the major concerns during review is the experiment design. The original paper lacks the results on real work dataset like ImageNet. Furthermore, the presentation is a little misleading. The authors addressed most of these problems in the revision.

Model compression and purring is a very important field for real world application, hence I choose to accept the paper.